# Exosome complex orchestrates developmental signaling to balance proliferation and differentiation during erythropoiesis

**Skye C McIver[1,2,3], Koichi R Katsumura[1,2,3], Elsa Davids[1,2,3], Peng Liu[3,4], Yoon-A Kang[1,2,3], David Yang[5], Emery H Bresnick[1,2,3]***

[1]Department of Cell and Regenerative Biology, University of Wisconsin School of Medicine and Public Health, Madison, United States; [2]UW-Madison Blood Research Program, University of Wisconsin School of Medicine and Public Health, Madison, United States; [3]Carbone Cancer Center, University of Wisconsin School of Medicine and Public Health, Madison, United States; [4]Department of Biostatistics and Medical Informatics, University of Wisconsin School of Medicine and Public Health, Madison, United States; [5]Department of Pathology, University of Wisconsin School of Medicine and Public Health, Madison, United States

**Abstract** Since the highly conserved exosome complex mediates the degradation and processing of multiple classes of RNAs, it almost certainly controls diverse biological processes. How this post-transcriptional RNA-regulatory machine impacts cell fate decisions and differentiation is poorly understood. Previously, we demonstrated that exosome complex subunits confer an erythroid maturation barricade, and the erythroid transcription factor GATA-1 dismantles the barricade by transcriptionally repressing the cognate genes. While dissecting requirements for the maturation barricade in *Mus musculus*, we discovered that the exosome complex is a vital determinant of a developmental signaling transition that dictates proliferation/amplification versus differentiation. Exosome complex integrity in erythroid precursor cells ensures Kit receptor tyrosine kinase expression and stem cell factor/Kit signaling, while preventing responsiveness to erythropoietin-instigated signals that promote differentiation. Functioning as a gatekeeper of this developmental signaling transition, the exosome complex controls the massive production of erythroid cells that ensures organismal survival in homeostatic and stress contexts.

*For correspondence: ehbresni@wisc.edu

**Competing interests:** The authors declare that no competing interests exist.

## Introduction

The highly conserved exosome complex, an RNA-degrading and processing machine, is expressed in all eukaryotic cells (*Januszyk and Lima, 2014*; *Kilchert et al., 2016*). The first subunit was discovered from an analysis of mechanisms controlling yeast rRNA synthesis. Mutations of *Exosc2 (Rrp4)* disrupt 5.8S rRNA 3'-end processing (*Mitchell et al., 1996*). Exosc2 assembles into a complex with components homologous to bacterial 3' to 5' exoribonuclease (PNPase) (*Mitchell et al., 1997*). Nine exosome complex subunits form a cylindrical core consisting of the RNA binding subunits Exosc1 (Csl4), Exosc2 (Rrp4) and Exosc3 (Rrp40), which cap a ring formed by Exosc4 (Rrp41), Exosc5 (Rrp46), Exosc6 (Mtr3), Exosc7 (Rrp42), Exosc8 (Rrp43) and Exosc9 (Rrp45) (*Liu et al., 2006*; *Makino et al., 2013*; *Makino et al., 2015*; *Wasmuth et al., 2014*) (*Figure 1A*). Despite homology with bacterial PNPases, the vertebrate core subunits lack RNA-degrading activity (*Liu et al., 2006*). Whereas Dis3 (Rrp44) (nuclear) and Dis3L (cytoplasmic) catalytic subunits bind the same position of

**eLife digest** Red blood cells supply an animal's tissues with the oxygen they need to survive. These cells circulate for a certain amount of time before they die. To replenish the red blood cells that are lost, first a protein called stem cell factor (SCF) instructs stem cells and precursor cells to proliferate, and a second protein, known as erythropoietin, then signals to these cells to differentiate into mature red blood cells. It is important to maintain this balance between these two processes because too much proliferation can lead to cancer while too much differentiation will exhaust the supply of stem cells.

Previous work has shown that a collection of proteins called the exosome complex can block steps leading towards mature red blood cells. The exosome complex controls several processes within cells by modifying or degrading a variety of messenger RNAs, the molecules that serve as intermediates between DNA and protein. However, it was not clear how the exosome complex sets up the differentiation block and whether it is somehow connected to the signaling from SCF and erythropoietin.

McIver et al. set out to address this issue by isolating precursor cells with the potential to become red blood cells from mouse fetal livers and experimentally reducing the levels of the exosome complex. The experiments showed that these cells were no longer able to respond when treated with SCF in culture, whereas the control cells responded as normal. Further experiments showed that cells with less of the exosome complex also made less of a protein named Kit. Normally, SCF interacts with Kit to instruct cells to multiply. Lastly, although the experimental cells could no longer respond to these proliferation signals, they could react to erythropoietin, which promotes differentiation. Thus, normal levels of the exosome complex keep the delicate balance between proliferation and differentiation, which is crucial to the development of red blood cells.

In future, it will be important to study the exosome complex in living mice and in human cells, and to see whether it also controls other signaling pathways. Furthermore, it is worth exploring whether this new knowledge can help efforts to produce red blood cells on an industrial scale, which could then be used to treat patients with conditions such as anemia.

the core complex, adjacent to Exosc4 and Exosc7, the predominantly nuclear catalytic subunit Exosc10 (Rrp6) binds the opposite site (*Dziembowski et al., 2007*; *Makino et al., 2015*) (*Figure 1A*). The catalytic subunits, which may function redundantly in certain contexts, mediate RNA degradation and/or processing (*Kilchert et al., 2016*). Unlike Dis3L and Exosc10, which are strictly exoribonucleases, Dis3 is also an endoribonuclease (*Lebreton et al., 2008*; *Tomecki and Dziembowski, 2010*). The core subunits, except Exosc1, are considered to confer structural integrity (*Liu et al., 2006*).

The apparent diversity of exosome complex-regulated RNAs (*Schneider et al., 2012*) suggests the complex controls a plethora of cellular processes. Exosome complex subunits regulate cell differentiation and are implicated in human pathologies. Exosc8, Exosc9 and Dis3 suppress erythroid maturation of primary murine erythroid precursor cells (*McIver et al., 2014*). EXOSC7, EXOSC9 and EXOSC10 maintain human epidermal progenitor function (*Mistry et al., 2012*). In principle, the exosome complex might control differentiation through complex RNA remodeling mechanisms or by targeting factors mediating the balance between self-renewal and lineage commitment, proliferation/amplification or terminal differentiation. *DIS3* has been implicated as a tumor suppressor mutated in multiple myeloma (*Chapman et al., 2011*) and is overexpressed in colorectal cancer (*de Groen et al., 2014*). *EXOSC8, EXOSC3 or EXOSC2* mutations cause syndromes with complex phenotypes including neurological defects, cerebellar hypoplasia, retinitis pigmentosum, progressive hearing loss and premature aging (*Boczonadi et al., 2014*; *Di Donato et al., 2016*; *Wan et al., 2012*).

As downregulating exosome complex subunits can yield compelling phenotypes, and exosome complex subunit mutations yield pathologies, it is instructive to consider whether the phenotypes reflect complex disruption or subunit-specific activities. By conferring exosome complex integrity, all exosome complex activities might require structural subunits, or sub-complexes might have distinct

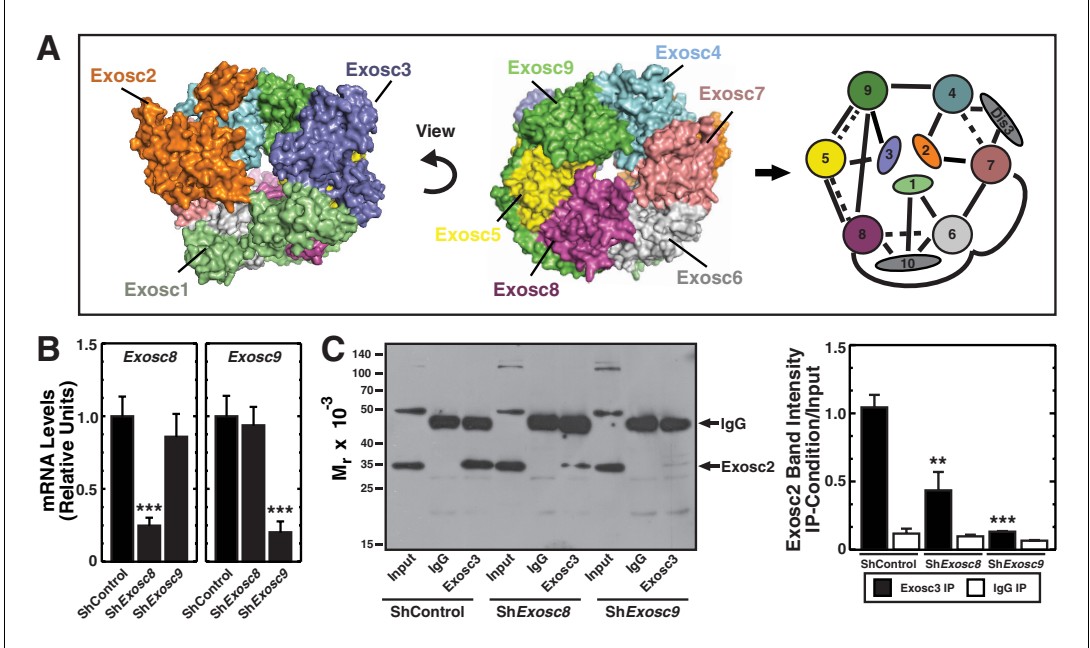

**Figure 1.** Exosc8 or Exosc9 downregulation disrupts protein-protein interactions within the exosome complex. (A) Crystal structure and model of the human exosome complex (*Liu et al., 2006*). Solid line, direct interactions; Dashed line, indirect interactions. (B) Real-time RT-PCR analysis of mRNA expression (mean ± SE, 3 independent replicates) in G1E-ER-GATA-1 cells 48 hr post-infection with either *Exosc8* or *Exosc9* shRNA expressing retrovirus. Values normalized to 18S expression and relative to the control. (C) Left: representative image of a semi-quantitative Western blot of Exosc2 co-immunoprecipitated with anti-Exosc3 antibody in G1E-ER-GATA-1 whole cell lysates prepared 48 hr post-*Exosc8* or *Exosc9* knockdown. Right: densitometric analysis of band intensity relative to the input for each knockdown condition (mean ± SE, 3 independent replicates). Statistical analysis of control and treatment conditions was conducted with the Student's T-test. *p<0.05, **p<0.01, ***p<0.001. Source data is available in *Figure 1—source data 1*.

The following source data and figure supplement are available for figure 1:

**Source data 1.** This Excel spreadsheet contains the values of each independent replicate for data presented as histograms (mean ± SE) in *Figure 1*.
**Figure supplement 1.** The RNA binding exosome complex component Exosc3 suppresses erythroid maturation.

functions (*Kiss and Andrulis, 2011*). The structural subunit requirement for complex stability in vitro (*Liu et al., 2006*), and lethality due to loss of structural components in yeast (*Allmang et al., 1999a*; *Allmang et al., 1999b*) support the importance of the intact complex. However, core subunit down-regulation revealed little overlap in the ensembles of regulated RNAs in *Drosophila* (*Kiss and Andrulis, 2010*) and differentially influenced RNA processing in humans (*Tomecki et al., 2010*). Moreover, *Arabidopsis Exosc1, Exosc2* and *Exosc4* knockouts yielded distinct phenotypes (*Chekanova et al., 2007*).

While investigating these models in the context of erythroid maturation, we discovered that Exosc8 or Exosc9 downregulation disrupted protein/protein interactions within the complex and greatly decreased expression of the receptor tyrosine kinase Kit. Loss of Stem Cell Factor (SCF)-induced Kit signaling occurred concomitant with precocious acquisition of erythropoietin signaling, which drives erythroid maturation. As Kit stimulates erythroid precursor cell proliferation, our results establish a paradigm in which the exosome complex regulates a receptor tyrosine kinase to orchestrate a vital developmental signaling transition dictating proliferation/amplification versus differentiation.

## Results

### Dismantling protein-protein interactions within the exosome complex

Previously, we demonstrated that downregulating exosome complex subunits (Exosc8, Exosc9 or Dis3) in murine fetal liver erythroid precursor cells induced erythroid maturation (*McIver et al., 2014*). Analogous to Exosc8 and Exosc9 (*McIver et al., 2014*), experiments in which Exosc3 expression is impaired suggest that this protein also suppresses maturation of primary murine fetal liver lineage-negative hematopoietic precursor cells. In particular, Exosc3 downregulation using two distinct shRNAs increased the R4 (late basophilic/orthochromatic erythroblasts) cell population nine fold (p=0.006 and p=0.01 for the two shRNAs, respectively) (*Figure 1—figure supplement 1*). However, it remains unclear whether the single subunit perturbations impact exosome complex integrity. To address this, we developed a co-immunoprecipitation assay to test whether individual components mediate complex integrity (*Figure 1A*). Using the X-ray crystal structure of the exosome complex as a guide (*Liu et al., 2006*), the strategy involved testing whether downregulating endogenous Exosc8 or Exosc9 alter interactions between endogenous Exosc2 and Exosc3, subunits that do not interact directly in the complex (*Figure 1A*). As Exosc2 and Exosc3 are only expected to co-immunoprecipitate when residing in the complex or a sub-complex, the extent of co-immunoprecipitation constitutes a metric of complex integrity.

We conducted the protein/protein interaction analysis in G1E cells stably expressing a conditionally active GATA-1 allele (G1E-ER-GATA-1), which mimic a normal erythroid precursor cell, the proerythroblast (*Weiss et al., 1997*). Estradiol activation of ER-GATA-1 induces an erythroid transcriptional program and recapitulates a physiological window of erythroid maturation (*Welch et al., 2004*). G1E-ER-GATA-1 cells were infected with retroviruses expressing control (luciferase) shRNA or shRNAs targeting *Exosc8* or *Exosc9*. Whole cell lysates prepared 48 hr post-infection were immunoprecipitated with anti-Exosc3 or isotype-matched control antibody, and Western blotting was conducted with anti-Exosc2 antibody. Whereas knocking down *Exosc8* or *Exosc9* by ~75% (*Figure 1B*) did not alter Exosc2 levels in the input, the knockdowns reduced the amount of Exosc2 recovered with the anti-Exosc3 antibody (*Figure 1C*). Densitometric analysis indicated that *Exosc8* and *Exosc9* knockdowns reduced the amount of Exosc2 co-immunoprecipitated with Exosc3 by 58 (p=0.007) and 87% (p=1.3 $\times$ 10$^{-4}$), respectively (*Figure 1C*, right). Of relevance to this result, yeast *Exosc8 (Rrp43)* mutations decrease exosome complex stability and RNA binding (*Lourenco et al., 2013*). As Exosc8 or Exosc9 downregulation disrupted Exosc3-Exosc2 interactions that only occur in the complex, erythroid maturation resulting from downregulating either of these subunits is associated with dismantling or destabilizing intra-complex protein-protein interactions.

### Exosome complex-regulated signaling transition dictates proliferation versus differentiation

Since Exosc8 or Exosc9 downregulation disrupts the exosome complex and promotes erythroid maturation, we investigated how integrity of the complex creates an erythroid maturation barricade. Although the parameters dictating the decision of whether an erythroid precursor cell undergoes sustained proliferation or differentiates into an erythrocyte are incompletely understood, cytokine signaling is a key determinant (*Lodish et al., 2010*). It is instructive to consider the relationship between the exosome complex-mediated erythroid maturation barricade and signaling mechanisms that orchestrate proliferation versus differentiation. In principle, the complex might enhance signaling that favors proliferation or oppose signaling mediating differentiation. Whereas SCF/Kit signaling supports hematopoietic stem/progenitor cell (HSPC) and erythroid precursor cell proliferation and survival (*Lennartsson and Ronnstrand, 2012*), erythropoietin (Epo)/Epo receptor signaling uniquely promotes terminal differentiation (*Munugalavadla and Kapur, 2005*; *Wu et al., 1995b*). Epo and SCF/Kit can synergistically promote proliferation and survival (*Joneja et al., 1997*; *Muta et al., 1994*; *Sui et al., 1998*; *Wu et al., 1997*).

To test whether the exosome complex and Epo signaling are functionally interconnected, murine fetal liver erythroid precursor cells were infected with control or *Exosc8* shRNA-expressing retroviruses. After culturing cells for 48 hr in media containing increasing amounts of Epo (0–0.5 U/ml), we conducted flow cytometric analysis with Annexin V and the membrane-impermeable dye DRAQ7 to quantitate live cells (DRAQ7⁻/Annexin V⁻), as well as late (DRAQ7⁺/Annexin V⁺) and early (DRAQ7⁻/

Annexin V[+]) apoptosis (*Figure 2A,B*). Without exogenous Epo, live cells decreased from 66 to 24% (control versus knockdown cells) (p=2 × 10$^{-5}$), and late apoptotic cells increased proportionally (24 to 67%, p=3 × 10$^{-4}$). Exosc8 downregulation increased late apoptosis two-fold (p=0.02) when cells were cultured in media containing 0.001 U/ml Epo. Exosc8 downregulation did not significantly influence the percentage of live and late apoptotic cells when cells were cultured with higher Epo concentrations. Surprisingly, downregulating Exosc8 rendered cell integrity hypersensitive to limiting concentrations of Epo.

As downregulating Exosc8 promotes erythroid maturation, the commencement and/or progression of maturation when Epo is limiting might be incompatible with cell survival. Alternatively, lowering Epo might impair a mechanism that supports erythroid precursor proliferation, independent of enhanced maturation; thus, precursor cell survival would be compromised. Using flow cytometry to quantitate the erythroid cell surface markers Ter119 and CD71, we determined the impact of Exosc8 downregulation on erythroid precursor cell maturation under normal and Epo-limiting conditions. In this expansion culture, control cells were largely unaffected by limiting Epo. Whereas Exosc8 downregulation stimulated erythroid maturation (1.5 fold increase in R2 to R3 transition, p=0.003) in media containing 0.5 U/ml Epo, maturation of Exosc8-knockdown cells did not proceed without

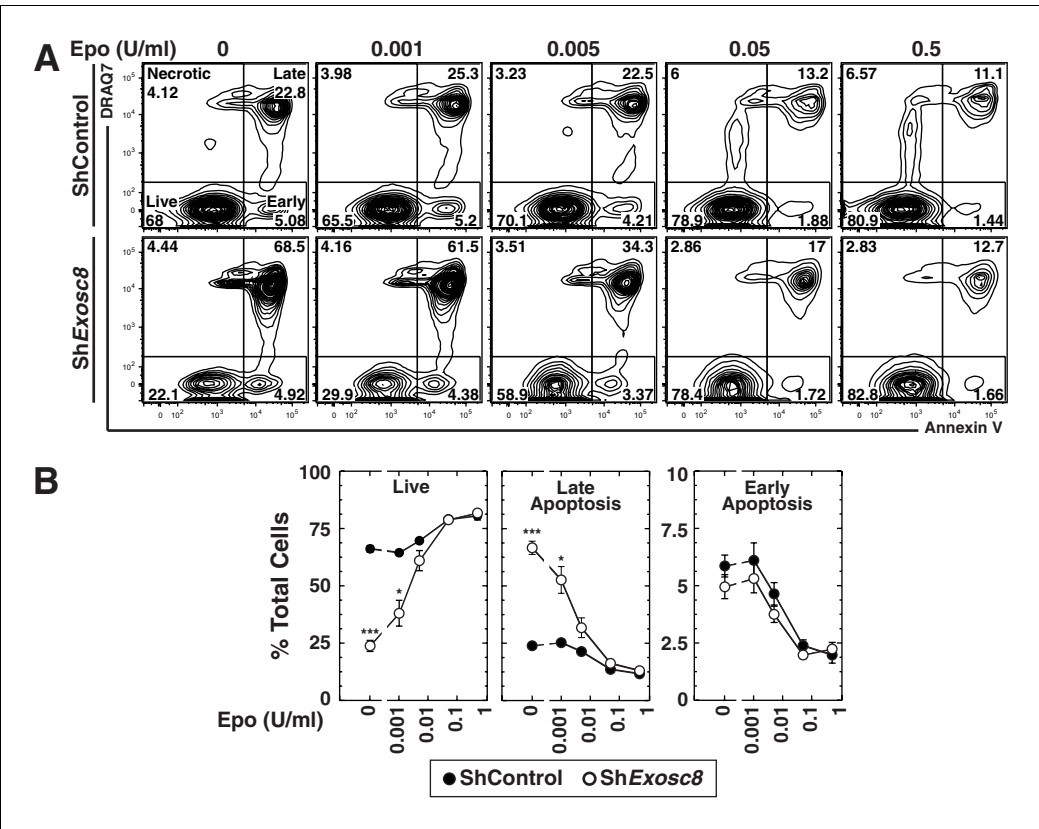

**Figure 2.** Exosome complex disruption renders primary erythroid cells hypersensitive to limiting erythropoietin concentrations. (**A**) Flow cytometric analysis with Annexin V and the membrane-impermeable dye DRAQ7 to quantitate apoptosis with control and *Exosc8*-knockdown primary erythroid cells expanded for 48 hr under Epo-limiting conditions. (**B**) Quantification of the percentage of primary erythroid cells in live, late and early apoptotic populations (mean ± SE, 4 biological replicates). Statistical analysis of control and treatment conditions was conducted with the Student's T-test. *p<0.05, **p<0.01, ***p<0.001. Source data is available in *Figure 2—source data 1*.

The following source data is available for figure 2:

**Source data 1.** This Excel spreadsheet contains the values of each biological replicate for data presented as line graphs (mean ± SE) in *Figure 2*.

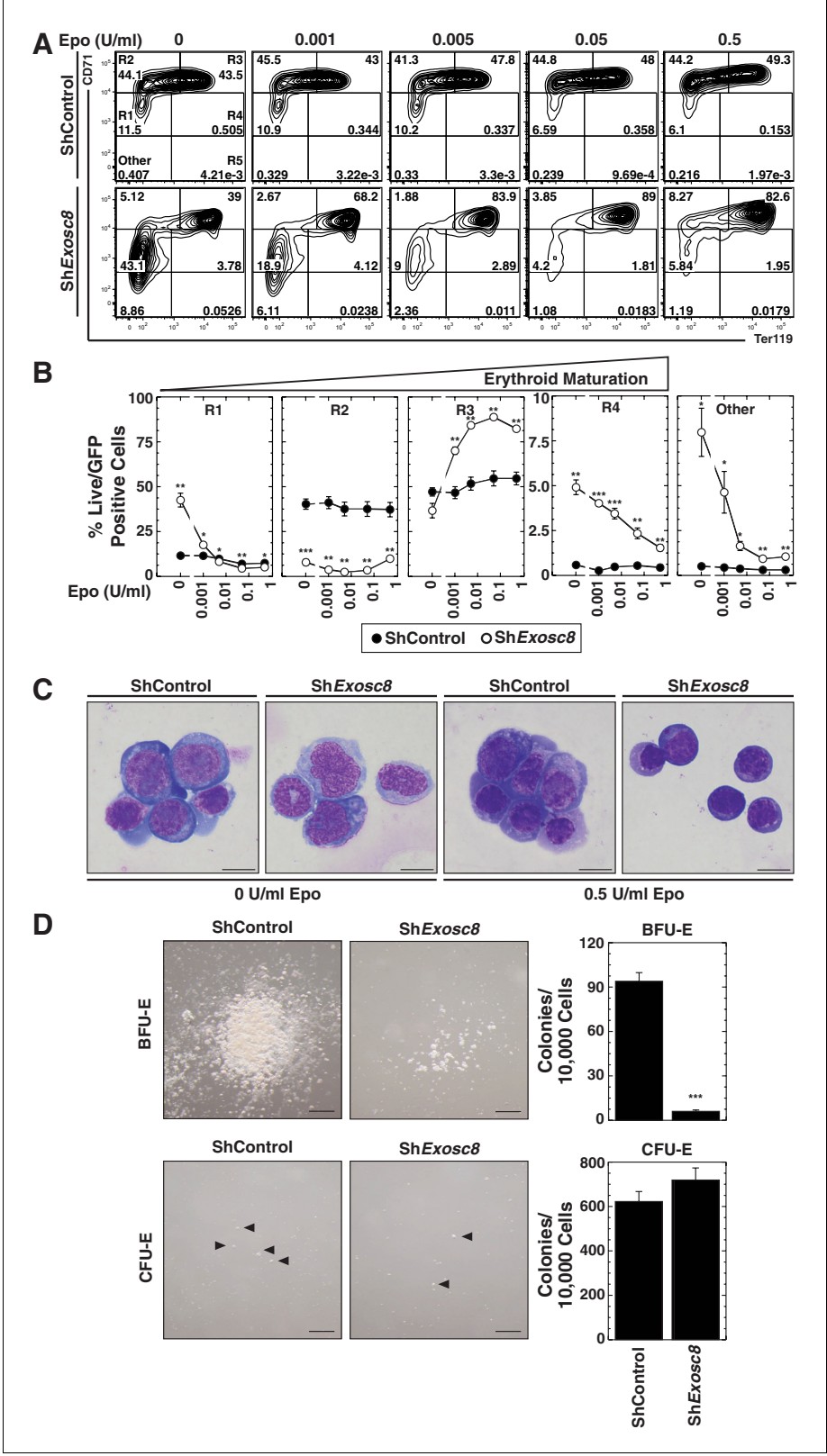

**Figure 3.** Erythropoietin is required for erythroid differentiation induced by disrupting the exosome complex. (**A**) Flow cytometric quantification of erythroid markers CD71 and Ter119 in live control and *Exosc8*-knockdown erythroid precursor cells cultured for 48 hr in Epo-limiting media. Representative plots with R1-R5 gates denoted. (**B**) Quantitation of the percentage of live cells in control and *Exosc8*-knockdown conditions from the R1-R4 and *Figure 3 continued on next page*

*Figure 3 continued*

non erythroid gates (mean ± SE, 4 biological replicates). (**C**) Representative images of Wright-Giemsa-stained, DRAQ7-negative erythroid precursor cells, infected with control or Sh*Exosc8* retrovirus. Cells were cultured with or without Epo for 48 hr (Scale bar, 10 μm). (**D**) Representative images (left) and quantitation (right) of erythroid colony forming unit activity with FACS-sorted R1 cells 24 hr after *Exosc8* knockdown (mean ± SE, 6 biological replicates) (Scale bar 200 μm). Statistical analysis of control and treatment conditions was conducted with the Student's T-test. *p<0.05, **p<0.01, ***p<0.001. Source data is available in *Figure 3—source data 1*.

The following source data and figure supplement are available for figure 3:

**Source data 1.** This Excel spreadsheet contains the values for each biological replicate for data presented as either line graphs or histograms (mean ± SE) in *Figure 3*.

**Figure supplement 1.** Analysis of the exosome complex-mediated erythroid maturation barricade using a distinct flow cytometric assay.

exogenous Epo (*Figure 3A,B*). Without Epo, Exosc8 downregulation also increased the percentage of immature erythroid precursors (R1). Morphological analysis of DRAQ7$^-$ cells using Giemsa stain provided further evidence for the Epo-dependent R2 to R3 transition resulting from Exosc8 knockdown (*Figure 3C*). The exosome complex-dependent erythroid maturation barricade was also quantitated using an alternative flow cytometric assay based on CD44 staining and cell size. Exosc8 or Exosc9 downregulation induced accumulation of more mature erythroid cells when cultured in 0.5 U/ml Epo (*Figure 3—figure supplement 1*). Orthochromatic erythroblasts (gate IV) increased 8 fold (p=0.006) or seven fold (p=3.5 × 10$^{-5}$) post-Exosc8 or -Exosc9 downregulation, respectively. Erythrocytes (gate VI) increased two fold (Exosc8, p=0.005, Exosc9, p=0.005). Immature erythroid cells (proerythroblasts and basophilic erythroblasts, gates I and II) decreased proportionally. Exosc8 downregulation reduced the number of early SCF, IL-3 and Epo-dependent BFU-E (Burst Forming Unit – Erythroid) colonies by 10 fold (p=4.7 × 10$^{-10}$). However, the number of Epo dependent CFU-E (Colony Forming Unit – Erythroid) colonies was unaffected (*Figure 3D*).

Erythroid maturation instigated by Exosc8 downregulation was Epo-dependent. We tested whether Exosc8 downregulation corrupted a proliferation/amplification mechanism, thereby promoting maturation. SCF signaling supports erythroid precursor proliferation (*Munugalavadla and Kapur, 2005*), whereas Epo signaling promotes maturation (*Wu et al., 1995b*). SCF and Epo can synergistically promote proliferation (*Joneja et al., 1997*; *Sui et al., 1998*; *Wu et al., 1997*). Using a phospho-flow cytometry assay (*Figure 4A*) (*Hewitt et al., 2015*), we quantitated the capacity of SCF (*Figure 4B*) or Epo (*Figure 4C*) to instigate cell signaling using the shared downstream substrate Akt. SCF induced maximal Akt phosphorylation in immature erythroblasts (Ter119$^-$/CD71$^{high}$) (5.5 fold, p=3 × 10$^{-6}$). As erythroid maturation progressed to Ter119$^+$/CD71$^{high}$ and Ter119$^+$/CD71$^{low}$ stages, the SCF response was diminished (*Figure 4B*). Exosc8 downregulation abrogated SCF-mediated induction of phospho-Akt (*Figure 4B*). Epo induced maximal Akt phosphorylation in Ter119$^+$/CD71$^{high}$ erythroblasts, and Exosc8 downregulation accelerated acquisition of this signaling response. Whereas Epo did not affect Akt phosphorylation in control Ter119$^-$/CD71$^{high}$ erythroblasts, Epo increased phospho-Akt 4 fold (p=0.003) in Exosc8-knockdown Ter119$^-$/CD71$^{high}$ erythroblasts (*Figure 4C*). Similar results were obtained using a phospho-flow cytometric assay to quantitate phosphorylation of ERK, an additional shared downstream effector of SCF and Epo signaling, although phospho-ERK was higher in the unstimulated Exosc8 condition in comparison with the control condition (*Figure 4—figure supplement 1*). Thus, downregulating an exosome complex subunit that dismantles intra-complex protein-protein interactions abrogates SCF signaling that supports precursor proliferation/amplification, while precociously inducing pro-differentiation Epo signaling. By orchestrating this developmental signaling transition, the exosome complex ensures a balance between proliferation/amplification and differentiation - a balance that shifts physiologically towards differentiation as GATA-1 represses genes encoding exosome complex subunits (*McIver et al., 2014*). Artificially, shRNA-mediated downregulation of exosome complex subunits, which impairs exosome complex integrity, skews the balance.

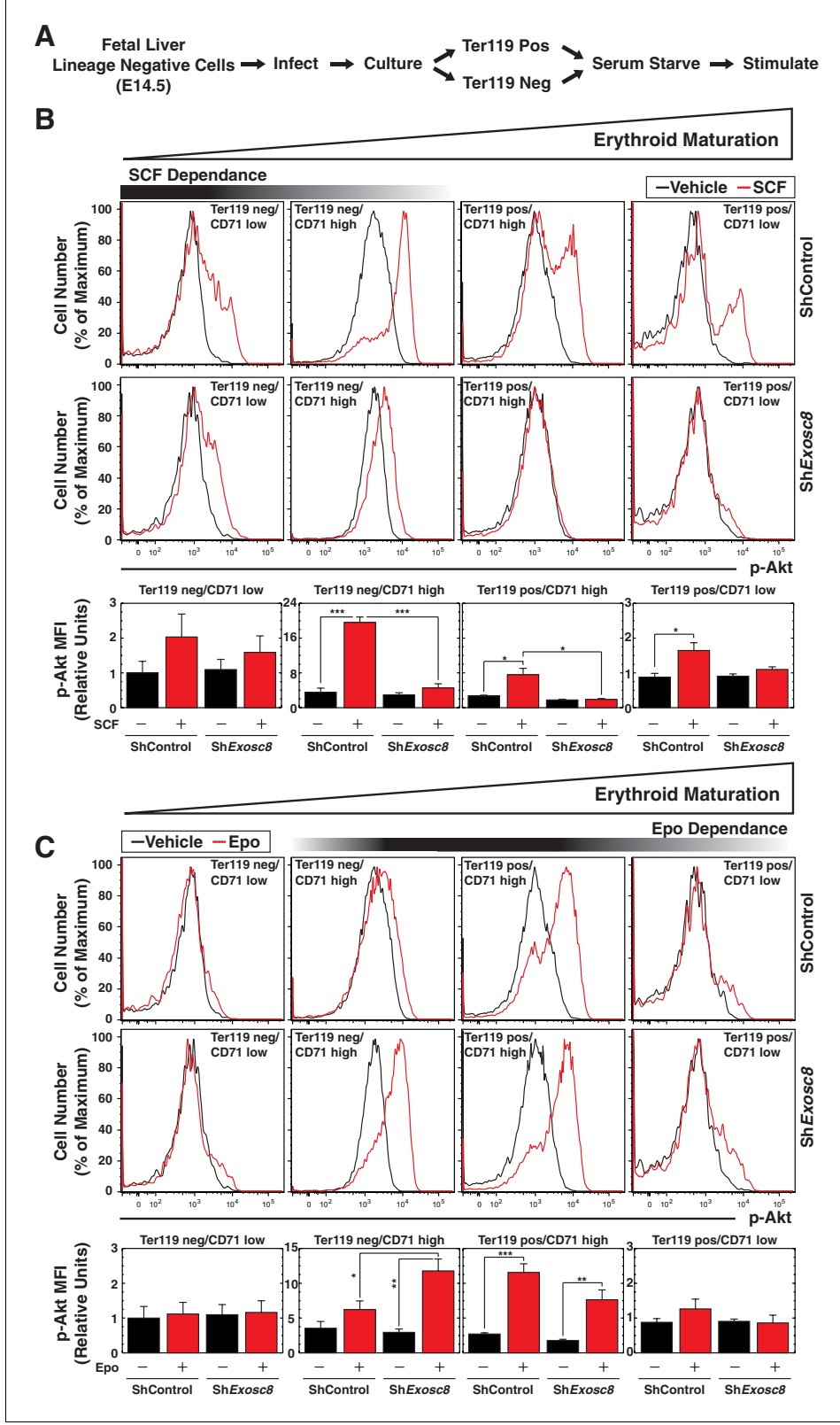

**Figure 4.** Exosome complex sustains proliferation signaling, while suppressing pro-differentiation signaling. (**A**) Experimental scheme: lineage-negative cells were isolated from E14.5 fetal livers, and infected with *luciferase* or *Exosc8* shRNAs. Cells were cultured for 48 hr and sorted into Ter119[+] and Ter119[-] populations using beads. After 1 hr of serum-starvation, cells were stimulated for 10 min with 10 ng/ml SCF or 2 U/ml Epo and fixed/

*Figure 4 continued on next page*

*Figure 4 continued*

permeabilized before staining for CD71 and p-Akt. (B) Top: p-Akt staining after stimulation with 10 ng/ml SCF in control and *Exosc8*-knockdown cells (6 biological replicates). Bottom: Relative p-Akt MFI after stimulation with 10 ng/ml SCF in control and *Exosc8*-knockdown cells. MFI expressed relative to unstimulated Ter119⁻/CD71^low control (mean ± SE, 6 biological replicates). (C) Top: p-Akt staining after stimulation with 2 U/ml EPO in control and *Exosc8*-knockdown cells (6 biological replicates). Bottom: Relative p-Akt MFI after stimulation with 2 U/ml Epo in control and *Exosc8*-knockdown cells. MFI expressed relative to unstimulated Ter119⁻/CD71^low control (mean ± SE, 6 biological replicates). ANOVA identified any significant variation within the experiment, and a Tukey-Kramer test identified the statistical relationship between each pair of samples. *p<0.05, **p<0.01, ***p<0.001. Source data is available in *Figure 4—source data 1*.

The following source data and figure supplement are available for figure 4:

**Source data 1.** This Excel spreadsheet contains the values for each biological replicate for data presented as histograms (mean ± SE) in *Figure 4*.

**Figure supplement 1.** Flow cytometric analysis of ERK phosphorylation reveals Exosc8 requirement to confer Kit signaling and to suppress Epo signaling.

## Mechanism governing exosome complex-regulated developmental signaling: transcriptional induction of an essential receptor tyrosine kinase

As downregulating Exosc8 abrogated SCF-induced Kit signaling, we tested whether Exosc8 is required for Kit expression in the erythroblast plasma membrane. Using flow cytometry with primary fetal liver erythroblasts, R2 erythroblasts expressed the highest levels of cell surface Kit, and Exosc8 downregulation reduced cell surface Kit to a nearly undetectable level (*Figure 5A*). Similarly, Exosc9 downregulation reduced cell surface Kit 6 fold in R2 cells (*Figure 5B*). We tested whether exosome complex disruption reduced *Kit* transcription, total Kit protein and/or Kit transit to the cell surface. Exosc8 downregulation reduced *Kit* mRNA and primary transcript levels at all stages of erythroid maturation (*Figure 5C*). By 24 hr post-infection with sh*Exosc8*-expressing retrovirus, total Kit protein declined 10 fold (p=0.046) in Ter119⁻ erythroid precursor cells (*Figure 5C*).

To further dissect the mechanism underlying exosome complex-dependent expression of Kit and establishment of SCF-induced Kit signaling, we tested whether exosome complex disruption influenced other genes downregulated during erythroid maturation. Since GATA-2 directly activates *Kit* transcription in immature erythroblasts, and GATA-1 directly represses *Kit* transcription during maturation (*Jing et al., 2008*; *Munugalavadla et al., 2005*), we tested whether exosome complex disruption impacted GATA-2 and GATA-1 levels. Exosc8 downregulation modestly increased *Gata2* mRNA in R3 cells, but GATA-2 protein levels were not significantly affected in Ter119⁻ cells (*Figure 5D*). Previously, we demonstrated that *Exosc8* knockdown did not influence *Gata1* mRNA levels (*McIver et al., 2014*). GATA-1 increased slightly in *Exosc8*-knockdown Ter119⁻ cells (1.4 fold, p=0.02) (*Figure 5D*). GATA-2 activates transcription of *Samd14*, encoding a facilitator of SCF/Kit signaling (*Hewitt et al., 2015*). In certain contexts, Kit signaling creates a positive autoregulatory loop that increases *Kit* transcription (*Zhu et al., 2011*). Downregulating Exosc8 had little to no effect on *Samd14* expression (*Figure 5D*). To assess whether Exosc8 downregulation reduced expression of genes resembling *Kit* in being transcriptionally downregulated upon maturation, we quantitated expression of *Vim* (encoding vimentin) (*DeVilbiss et al., 2015*) and the GATA-2 target gene *Hdc* (encoding histidine decarboxylase) (*Katsumura et al., 2014*). Downregulating Exosc8 did not affect the maturation-dependent reduction in *Vim* or *Hdc* mRNA levels (*Figure 5D*). The Exosc8 requirement for *Kit* transcription and expression of functional, cell surface Kit therefore does not involve a major alteration in GATA factor levels nor a general Exosc8 activity to sustain expression of genes destined for downregulation upon erythroid maturation.

Previously, we demonstrated that exosome complex disruption induces erythroid precursor cells to arrest in the G1 phase of the cell cycle and is associated with increased expression of genes promoting (or implicated in promoting) cell cycle arrest (*McIver et al., 2014*). As erythroid maturation requires cell cycle progression (*Pop et al., 2010*), we asked whether Kit downregulation precedes, is

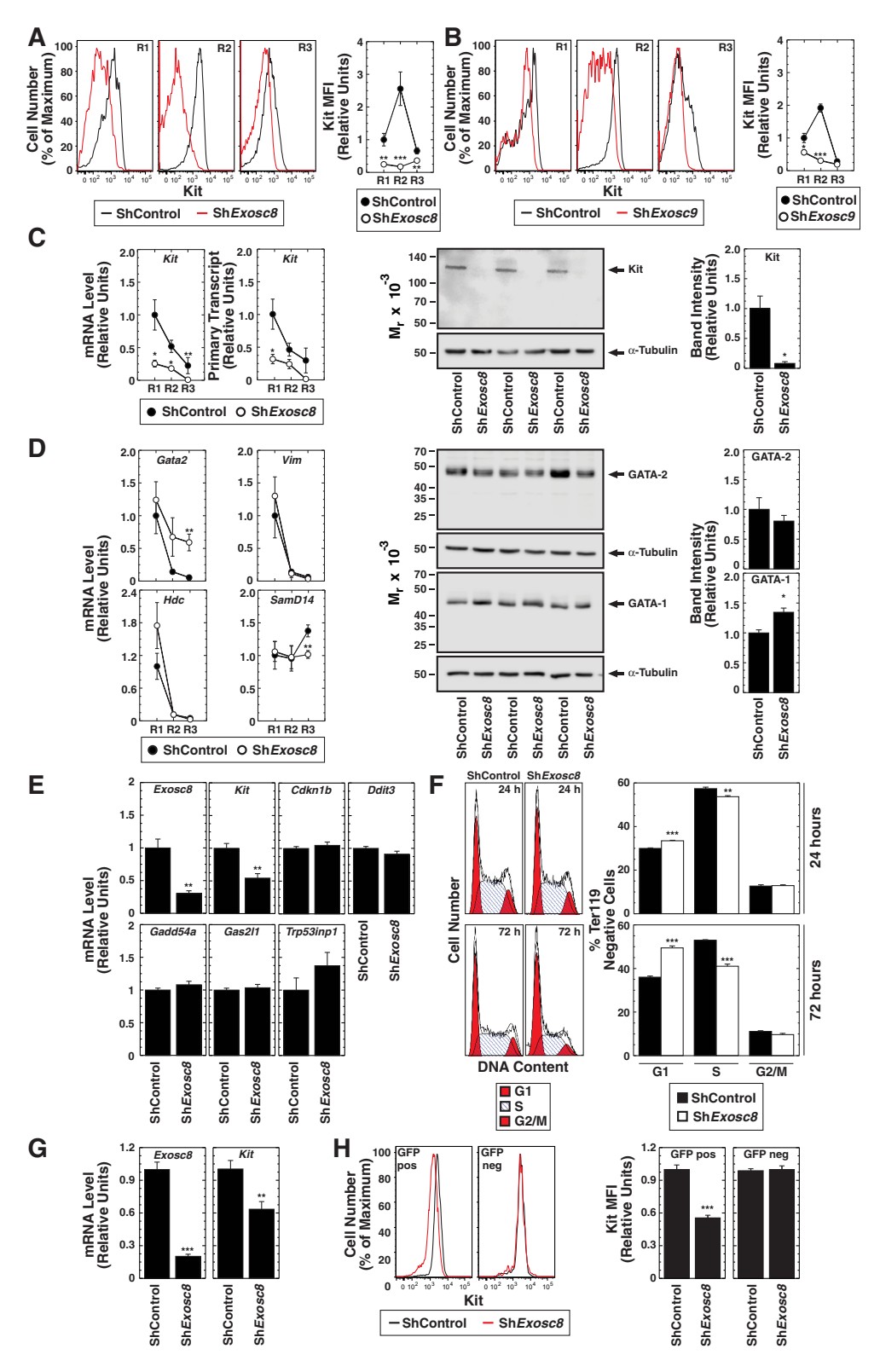

**Figure 5.** Exosome complex requirement for Kit expression. (**A**) Left: surface Kit in R1-R3 populations 48 hr post-*Exosc8* knockdown. Representative plots. Right: surface Kit MFI relative to control R1 (mean ± SE, 5 biological replicates). (**B**) Left: surface Kit in R1-R3 cells 48 hr post-*Exosc9* knockdown. Representative plots. Right: surface Kit MFI relative to control R1 (mean ± SE, 8 biological replicates). (**C**) Left: real-time RT-PCR of *Kit* mRNA and primary transcripts in sorted R1-R3 populations 72 hr post-infection with shControl or sh*Exosc8* normalized to 18S and relative to control R1 (mean ± SE, *Figure 5 continued on next page*

*Figure 5 continued*

6 biological replicates). Middle: Kit Western blot with Ter119⁻ cells 24 hr post-infection (mean ± SE, 3 biological replicates). (D) Left: real-time RT-PCR of erythroid mRNAs in sorted R1-R3 populations 72 hr post-infection with shControl or shExosc8 (mean ± SE, 6 biological replicates). Middle: GATA-2 and GATA-1 Western blot with Ter119⁻ cells 24 hr post-infection. Right: densitometric analysis normalized to tubulin and relative to shControl (mean ± SE, 3 biological replicates). (E) qRT-PCR of *Exosc8* and *Kit* mRNA and GATA-1/Exosc8-regulated cell cycle arrest genes in primary erythroid precursor cells 24 hr post-infection. Normalized to 18S and relative to the control (mean ± SE, 5 biological replicates). (F) Cell cycle analysis of control and Exosc8-knockdown Ter119⁻ cells 24 (top) and 72 hr (bottom) post-infection (mean ± SE, 6 biological replicates) (G) qRT-PCR analysis of *Exosc8* and *Kit* mRNA in G1E cells 48 hr post-infection with shControl or shExosc8 retrovirus, normalized to 18S and expressed relative to the control (mean ± SE, 3 independent experiments) (H) Cell surface Kit expression in infected (GFP⁺) and uninfected (GFP⁻) populations of G1E cells 48 hr post-infection with shExosc8 (mean ± SE, 3 independent experiments). Statistical analysis of control and treatment conditions was conducted with the Student's T-test *p<0.05, **p<0.01, ***p<0.001. Source data is available in *Figure 5—source data 1*.
The following source data is available for figure 5:

**Source data 1.** This Excel spreadsheet contains the values for each biological replicate for data presented as either line graphs or histograms (mean ± SE) in *Figure 5*.

concomitant with, or a consequence of GATA-1 and Exosc8-mediated regulation of cell cycle arrest genes. At 24 hr post-infection of fetal liver erythroid precursor cells with shRNA-expressing retrovirus, *Exosc8* mRNA declined by 70% (p=0.005). Whereas *Kit* mRNA decreased by 46% (p=0.002), expression of the GATA-1/Exosc8-regulated cell cycle-regulatory genes *Cdkn1b* (p27Kip1), *Ddit3*, *Gadd45a*, *Gas2l1* and *Trp53inp1* was not significantly altered (*Figure 5E*). Analysis of the cell cycle status of Kit-expressing Ter119⁻ erythroid precursor cells 24 and 72 hr post-infection revealed only a slight increase in the percentage of G1 cells at 24 hr (30% in control and 33.5% in *Exosc8*-knockdown cells (p=2 × 10⁻⁶)). At 72 hr post-infection, 49.5% of *Exosc8*-knockdown Ter119⁻ cells were in G1, in comparison with 36% of control cells (p=1.6 × 10⁻⁶) (*Figure 5F*). Thus, the prominent cell cycle arrest resulting from Exosc8 downregulation occurs after Kit is repressed. To more definitively establish the relationship between Kit downregulation and cell cycle arrest, we tested whether Exosc8 downregulation reduces Kit expression in a system not competent for differentiation. shRNA-mediated *Exosc8* downregulation in GATA-1-null G1E proerythroblast-like cells reduced *Kit* mRNA by 37% (p=0.002) (*Figure 5G*), and Kit cell surface expression by 44% in the GFP⁺ population (p=3.3 × 10⁻⁸). Importantly, cell surface Kit was unaffected in the GFP⁻ population (*Figure 5H*). In aggregate, these analyses support a model in which exosome complex disruption downregulates Kit expression independent of alterations in cell cycle status and is not a consequence of cell cycle arrest or cellular maturation. Consistent with these results, mining RNA-seq data from normal and *Exosc3⁻/⁻* ES cells (*Pefanis et al., 2015*) revealed *Kit* expression 5.9 fold lower in mutant versus control cells (transcripts per million, false discovery rate < 0.05).

We tested whether the Exosc8 requirement for *Kit* primary transcript, mRNA and protein expression involved alterations in the distribution of transcriptionally-competent serine 5-phosphorylated RNA polymerase II (Pol II) at *Kit*. Using quantitative ChIP analysis with control and *Exosc8*-knockdown Ter119⁻ cells, Exosc8 downregulation reduced phospho-Ser5 Pol II occupancy within the coding region (+5 kb) and 3′ UTR, but not at the promoter (*Figure 5E*). Exosc8 downregulation did not alter phospho-Ser5 Pol II occupancy at the active *Rpb1* gene or the inactive *Krt5* gene (*Figure 6A*). These results provide further evidence that exosome complex-mediated *Kit* expression involves a transcriptional mechanism. As GATA-1 levels increased slightly after Exosc8 downregulation in Ter119⁻ cells, and GATA-1 represses *Kit* transcription, we tested whether GATA-1 occupancy at *Kit* was altered. Exosc8 downregulation did not influence GATA-1 occupancy at -114, +5 and +58 kb sites, relative to the promoter (*Figure 6A*). The nearly complete *Kit* repression upon Exosc8 downregulation did not involve detectable changes in GATA-1 occupancy.

In *Drosophila*, a genome-wide analysis of Exosc10 (Rrp6) and Exosc3 (Rrp40) chromatin occupancy revealed occupancy predominantly at chromatin insulators and promoters of active genes (*Lim et al., 2013*). However, considering that the exosome complex associates with elongating RNA polymerase II (*Andrulis et al., 2002*) and rapidly degrades promoter upstream transcripts (PROMPTs), which are believed to be generated at many genes (*Lubas et al., 2011*; *Preker et al., 2011*; *Preker et al., 2008*), one might predict that the exosome complex has a broader distribution

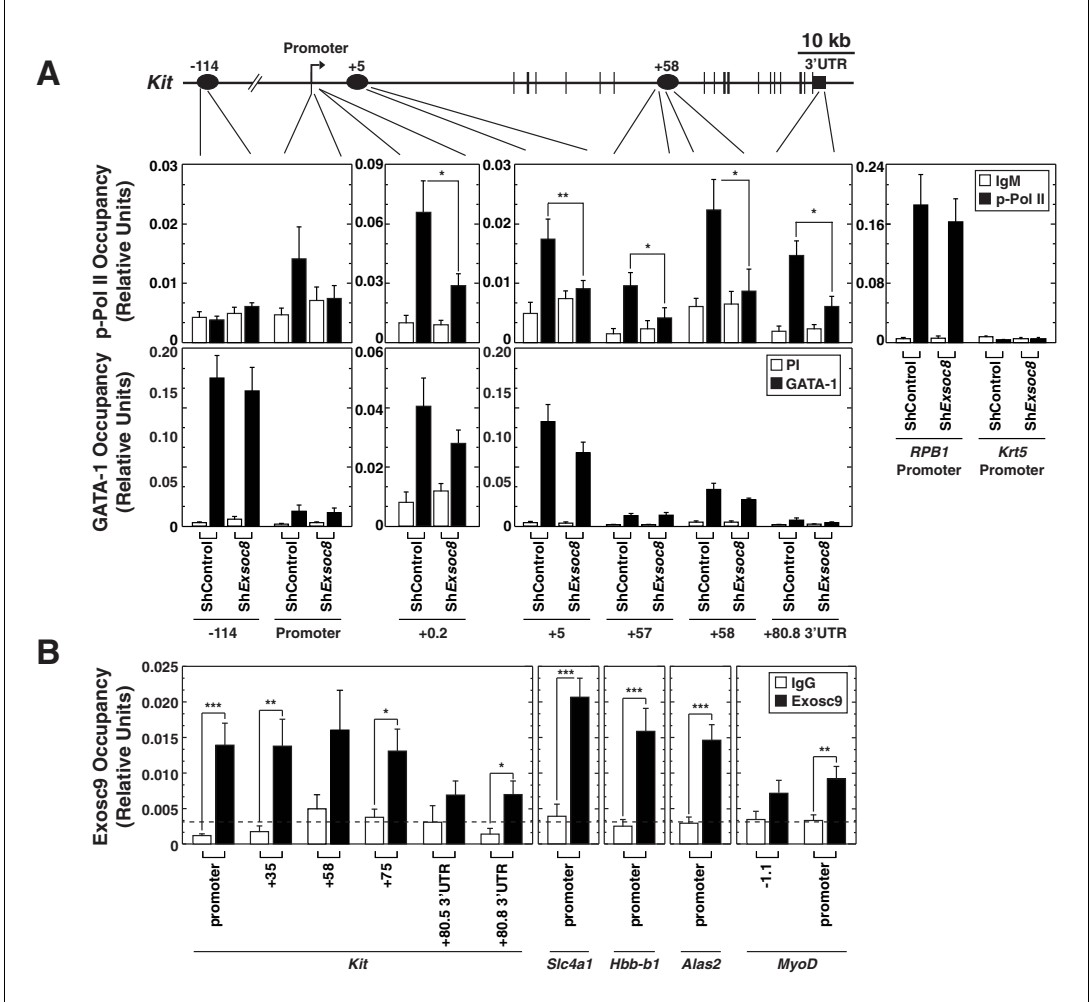

**Figure 6.** Exosome complex occupies the *Kit* locus and is required for active RNA Polymerase II occupancy at *Kit*. (**A**) qChIP of serine 5-phospho Pol II, and GATA-1 occupancy at *Kit* in control and Exosc8-knockdown Ter119⁻ erythroid precursor cells 24 hr post-infection (mean ± SE, 6 independent experiments). (**B**) qChIP of Exosc9 occupancy at *Kit* and promoters of other exosome complex-regulated erythroid genes (*Alas2, Hbb-b1* and *Slc4a1*) in erythroid precursor cells after culturing for 48 hr (mean ± SE, 3 biological replicates). Statistical analysis of control and treatment conditions was conducted with the Student's T-test. *p<0.05, **p<0.01, ***p<0.001. Source data is available in *Figure 6—source data 1*.

The following source data is available for figure 6:

**Source data 1.** This Excel spreadsheet contains the values for each biological replicate for data presented in histograms (mean ± SE) in *Figure 6*.

in and surrounding genes. To investigate the mechanism by which the exosome complex confers *Kit* transcription and suppresses *Alas2, Hbb-b1* and *Slc4a1* transcription (*McIver et al., 2014*), we asked whether the exosome complex occupies these loci. Quantitative ChIP analysis with fetal liver erythroid precursor cells expanded for 48 hr revealed endogenous Exosc9 occupancy at the *Kit* promoter, the coding region and the 3'-UTR, with occupancy higher at the promoter and coding region sites versus the 3'-UTR (*Figure 6B*). Exosc9 also occupied the *Alas2, Slc4a1* and *Hbb-b1* promoters, with less occupancy at the *MyoD* promoter (*Figure 6B*). Exosc9 occupancy implies that the exosome complex directly regulates transcription of these loci.

Although Kit downregulation correlated with erythroid maturation induced by exosome complex disruption, whether this reflects causation was unclear. To establish whether Kit downregulation is required for erythroid maturation induced by exosome complex disruption, we asked whether enforced Kit expression in Exosc8-knockdown Ter119⁻ cells opposed differentiation. At 24 hr post-infection, there was little to no difference in the maturation state of control and Exosc8-knockdown

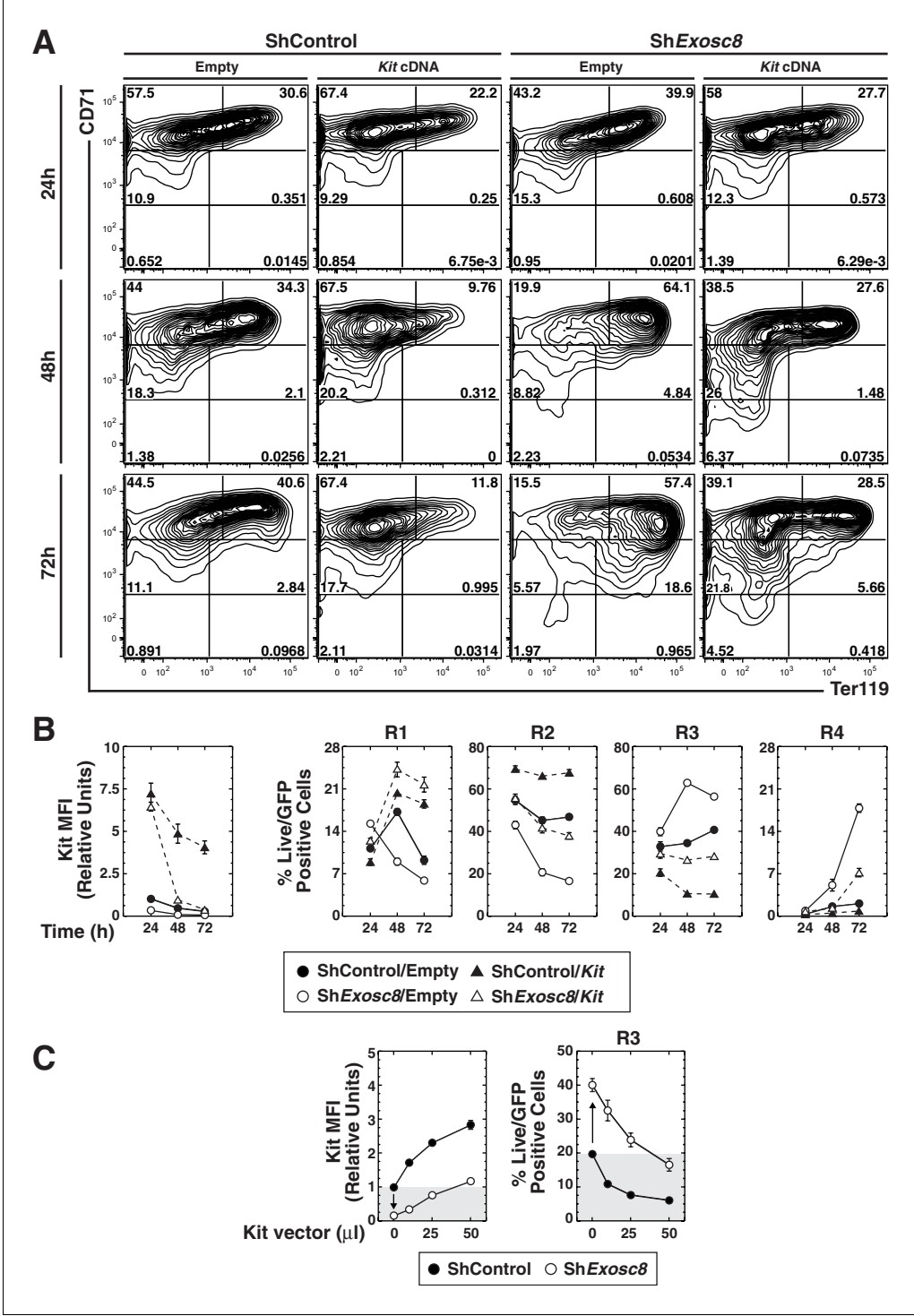

**Figure 7.** Functional link between Kit downregulation and erythroid differentiation induced by disrupting the exosome complex. (**A**) Erythroid maturation analyzed by flow cytometric quantitation of CD71 and Ter119 post-*Exosc8* knockdown and/or Kit expression in primary erythroid precursor cells expanded for 72 hr. Representative flow cytometry plots, with the R1-R5 gates denoted. (**B**) Left: relative Kit MFI post-*Exosc8* knockdown and/or Kit overexpression (mean ± SE, 4 biological replicates). Right: percentage of primary erythroid precursor cells in the R1-R4 gates (mean ± SE, 4 biological replicates). (**C**) Left: relative Kit MFI 48 hr post-*Exosc8* knockdown in cells infected with increasing amounts of a Kit-expressing retrovirus. The arrow depicts Kit downregulation resulting from knocking-down Exosc8. Right: percentage of erythroid precursor cells in the R3 population 48 hr post-infection with sh*Exosc8* in cells infected with increasing amounts of Kit-expressing retrovirus. The arrow depicts the

*Figure 7 continued on next page*

*Figure 7 continued*

increased R3 population post-Exosc8 knockdown. ANOVA identified any significant variation between experimental groups then a Tukey-Kramer test identified the statistical relationship between each pair of samples, *p<0.05, **p<0.01, ***p<0.001. Source data is available in *Figure 7—source data 1*

The following source data is available for figure 7:

**Source data 1.** This Excel spreadsheet contains the values for each biological replicate for data presented in line graphs (mean ± SE) in *Figure 7*.

cells (*Figure 7A,B*). By 48 hr, Exosc8 downregulation increased R3 cells from 35 to 63% (p=4 × 10$^{-7}$), while reducing R2 cells two-fold (p=4 × 10$^{-5}$). Kit expression prevented Exosc8 knockdown-dependent maturation, reducing R3 cells from 63 to 27% (p=1 × 10$^{-7}$). Kit expression, concomitant with Exosc8 downregulation, rescued R2 cells (increased from 21 to 42%, p=8 × 10$^{-4}$) to a level indistinguishable from control cells (45%). At 48 hr, Kit expression in control cells increased R2 cells from 45 to 66% (p=6 × 10$^{-5}$). At 72 hr, Exosc8-knockdown cells matured further (18% R4, 56% R3 and 17% R2 versus 2% R4 (p=4 × 10$^{-5}$), 41% R3 (p=3 × 10$^{-5}$), and 47% R2 (p=3 × 10$^{-6}$) for control cells). Expressing Kit in Exosc8-knockdown cells reduced R4 from 18 to 7% (p=4 × 10$^{-5}$), R3 from 56 to 28% (p=9 × 10$^{-7}$), while increasing R2 from 17 to 38% (p=5 × 10$^{-5}$). At 72 hr, Kit expression increased R2 cells from 47 to 68% (p=6 × 10$^{-5}$).

We evaluated the relationship between the level of Kit required to oppose erythroid maturation, caused by exosome complex disruption, and endogenous Kit expression. Cells were infected with a range of retrovirus concentrations to establish the amount required to restore Kit to the endogenous level after Exosc8 downregulation. Exosc8 downregulation reduced Kit MFI six fold (p=0.001). At 48 hr post-infection, 50 μl of Kit retroviral supernatant (half of that used in the time-course) increased Kit cell surface expression in Exosc8-knockdown cells to approximately the endogenous level (1.2 fold higher than control). Under these conditions, infection of control cells increased Kit cell surface expression three fold (p=0.002). Exosc8 downregulation, without ectopic Kit expression, increased the percentage of R3 cells ~two fold (p=0.003) 48 hr post-infection. The increased R3 population cells induced by Exosc8 downregulation was prevented by enforced Kit expression at 1.2 fold higher than the endogenous level (control R3, 19.7% versus Exosc8/Kit R3, 16.5%). A three fold increase in Kit cell surface expression in the control condition decreased the R3 population cells from 19.7% to 6% (p=0.001) (*Figure 7C*). As enforced Kit expression negated erythroid maturation induced by exosome complex disruption, the exosome complex activity to establish and/or maintain Kit expression is a critical step in the exosome complex-dependent maturation process.

## Discussion

Regulating the proliferation and differentiation balance of stem and progenitor cells through intrinsic and extrinsic mechanisms ensures normal development and physiology. Whereas exaggerated proliferation of stem and/or progenitor cells can underlie cancer (*He et al., 2004*), a differentiation bias can exhaust the precursor cells. As the intestinal epithelial layer is replaced every 4-5 days, the proliferation versus differentiation balance of intestinal stem cells must be exquisitely regulated (*Clevers et al., 2014*). Wnt signaling promotes intestinal stem cell self-renewal, while Bone Morphogenetic Protein (BMP) signaling suppresses self-renewal and promotes differentiation (*Clevers et al., 2014*; *He et al., 2004*).

In hematopoiesis, SCF signaling promotes erythroid precursor cell proliferation at the expense of differentiation (*Haas et al., 2015*; *Muta et al., 1995*). Epo provides a pro-differentiation stimulus (*Wu et al., 1995b*). During stress erythropoiesis, Epo, SCF and glucocorticoids synergistically promote erythroid cell expansion (*Kolbus et al., 2003*; *Wessely et al., 1997*) and Kit downregulation promotes maturation (*Haas et al., 2015*; *Munugalavadla et al., 2005*). Epo and Kit can synergize (*Joneja et al., 1997*; *Sui et al., 1998*; *Wu et al., 1995a*) or be antagonistic (*Haas et al., 2015*; *Joneja et al., 1997*; *Kosmider et al., 2009*). Kit signaling induces Epo receptor tyrosine phosphorylation (*Wu et al., 1995a*), although synergistic enhancement of proliferation may involve convergence of SCF and Epo signals on ERK1/2 via distinct pathways (*Sui et al., 1998*). Constitutively

active Kit inhibits Epo-mediated Akt phosphorylation, increasing apoptosis in mature erythroid cells (*Haas et al., 2015*). GATA-1 and the cooperating transcription factor Scl/TAL1 repress *Kit* transcription as erythroblasts acquire Epo-dependence (*Munugalavadla et al., 2005*; *Vitelli et al., 2000*).

While multiple pathways are implicated in controlling the proliferation and differentiation balance of stem and progenitor cells, many questions remain regarding how pathways intersect or function in parallel. Herein, we demonstrated that the exosome complex is a critical determinant of the proliferation versus differentiation balance of erythroid precursor cells, and many unanswered questions existed regarding how this balance is controlled (*Figure 8*). The exosome complex establishes the balance by ensuring expression of a receptor tyrosine kinase, Kit, which is essential for proliferation, under conditions in which progenitors are not competent to transduce Epo pro-differentiation signals. Disrupting exosome complex integrity downregulated Kit and SCF-mediated proliferation signaling, while inducing Epo signaling (*Figure 8*). Physiologically, GATA-1 represses expression of exosome complex subunits, instigating the developmental signaling transition. Reduced SCF signaling upon exosome complex disruption would be predicted to underlie or contribute to activation of Epo signaling in erythroid precursor cells (*Figures 4*, *5*) and promote erythroid differentiation (*Figures 3*, *7*). Our results establish a paradigm in which an RNA-regulatory machine orchestrates the balance between opposing developmental signaling pathways. Considering the apparent ubiquitous expression of the exosome complex, it is attractive to propose that the paradigm may operate to control additional progenitor cell transitions.

Mice homozygous for the W mutation (white spotting) in *Kit* die perinatally from severe macrocytic anemia (*Bernstein et al., 1990*; *Waskow et al., 2004*). The anemia likely reflects the *Kit* requirement for HSPC genesis, maintenance and function (*Bernstein et al., 1990*; *Ding et al., 2012*). Our analyses with primary fetal liver hematopoietic precursors cultured under conditions that support erythroid precursor growth and differentiation demonstrated that Exosc8 downregulation reduces *Kit* expression and signaling in erythroid precursors. These findings would not have been predicted from prior work with W mutant multipotent hematopoietic precursors in vivo, given the impact of Kit signaling on multiple cell types.

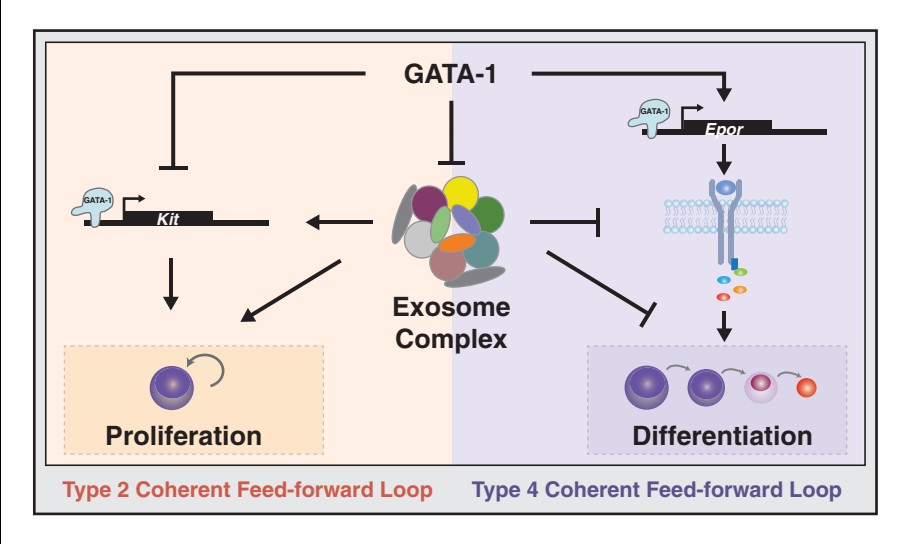

**Figure 8.** Exosome complex function to orchestrate developmental signaling pathways that control proliferation versus differentiation. The master regulator of erythropoiesis GATA-1 represses *Kit* transcription and upregulates *EpoR* transcription, thus establishing the developmental signaling circuitry for erythroid maturation. GATA-1 represses genes encoding exosome complex subunits, which promotes erythroid maturation. The exosome complex confers *Kit* expression and establishes competence for SCF-induced Kit signaling. Disruption of this mechanism abrogates Kit signaling and instigates Epo signaling, which favors erythroid precursor maturation versus self-renewal.

Exosome complex functions have expanded considerably since the discovery of its role in rRNA maturation (*Mitchell et al., 1996*) to include coding and non-coding RNA degradation and processing (*Kilchert et al., 2016*; *Schneider et al., 2012*). The exosome complex is enriched at actively transcribed genes (*Andrulis et al., 2002*; *Lim et al., 2013*), regulates transcription start site usage (*Jenks et al., 2008*; *Kuehner and Brow, 2008*), influences heterochromatin and post-transcriptional gene silencing (*Eberle et al., 2015*) and regulates superenhancer activity (*Pefanis et al., 2015*). Gene promoters are bidirectional in nature (*Andersson et al., 2015*), and the exosome complex rapidly degrades PROMPTs (*Lubas et al., 2011*; *Preker et al., 2011*; *Preker et al., 2008*). Depletion of exosome complex components stabilizes PROMPTs and additional cryptic transcripts present at gene loci, including PSARs (promoter-associated small RNAs), TSARs (terminator-associated small RNAs), PALRs (promoter-associated long RNAs), TSSa RNAs (transcription start site-associated RNAs) and eRNAs (enhancer RNAs) (*Jensen et al., 2013*; *Lubas et al., 2015*; *Pefanis et al., 2015*). The exosome complex is also implicated in transcriptional regulation, and since regulatory RNAs can be integral components of transcriptional mechanisms (*Belostotsky, 2009*; *Jensen et al., 2013*), this can be explained, in part, through RNA-regulatory activity. While knowledge of molecular mechanisms underlying exosome complex function are sophisticated, many questions remain unanswered regarding its functions in distinct cellular contexts during development, physiology and pathologies.

Genes encoding exosome complex components are linked to pathologies including cancer (*DIS3*) (*Chapman et al., 2011*), immune disorders (*EXOSC9* and *EXOSC10*) (*Allmang et al., 1999b*) and congenital neurological disorders (*EXOSC3* and *EXOSC8*) (*Boczonadi et al., 2014*; *Rudnik-Schoneborn et al., 2013*; *Wan et al., 2012*). Despite these human disease links, the apparent diversity of RNA targets and multiple functions of the exosome complex, little is known about its role in cell fate decisions. Downregulating *EXOSC10, EXOSC9* or *EXOSC7* expression caused differentiation of human epididymal progenitor cells. Mechanistically, the exosome complex represses *GRHL3*, a key transcription factor promoting epidermal differentiation (*Mistry et al., 2012*).

In summary, we have described the fundamental importance of the exosome complex to orchestrate a critical developmental signaling transition that determines whether erythroid precursors differentiate. Given the importance of ascribing biological functions for regulatory RNAs, the machinery mediating their biosynthesis and exosome complex functions to process diverse RNAs, it will be instructive to identify exosome complex-regulated RNA ensembles mediating cell fate transitions. Furthermore, we anticipate that the results described herein can be leveraged to improve strategies for the industrial-level generation of erythroid cells to achieve therapeutic goals. Finally, considering the common occurrence of Kit-activating mutations in cancers (*Lennartsson and Ronnstrand, 2006*), it will be of considerable interest to analyze the paradigm described herein in physiological states in vivo, to determine if it can be extrapolated to cancer, and to establish whether the exosome complex constitutes a promising therapeutic target for Kit-driven pathologies.

## Materials and methods

### Generation of 3D exosome complex structures

Protein structure coordinate files for the human exosome complex (*Liu et al., 2006*) were downloaded from the Research Collaboratory for Structural Bioinformatics Protein Data Bank (www.RCSB.org, accession number 2NN6). Images were generated using PyMOL (www.PyMOL.org, Schrödinger, New York, NY).

### Primary erythroid precursor cell isolation

Primary erythroid precursors were isolated from E14.5 mouse C57BL/6 fetal livers using EasySep negative selection Mouse Hematopoietic Progenitor Cell Enrichment Kit (StemCell Technologies, Vancouver, Canada) as described (*McIver et al., 2014*).

### Cell culture

G1E-ER-GATA-1 (RRID:CVCL_D047) cells were cultured in Iscove's Modified Dulbecco's Medium (IMDM) (ThermoFisher, Waltham, MA) containing 15% FBS (Gemini, West Sacramento, CA), 1% antibiotic/antimycotic (Corning, Tewksbury, MA), 2 U/ml erythropoietin (Amgen, Thousand Oaks, CA), 120 nM monothioglycerol (Sigma, St Louis, MO), 0.6% conditioned medium from an SCF producing

CHO cell line, and 1 µg/ml puromycin (Gemini) (*Fujiwara et al., 2009*). G1E (RRID:CVCL_D046 cells were cultured without puromycin.

G1E cells were derived from GATA-1-null murine ES cells. ES cells were cultured under conditions that promoted the development of definitive erythroid cells (*Weiss et al., 1997*). G1E-ER-GATA-1 cells are G1E cells stably expressing GATA-1 fused to the ligand-binding domain of human estrogen receptor (*Gregory et al., 1999*). G1E and G1E-ER-GATA-1 cells were a kind gift from Dr. Mitchell J. Weiss (St. Judes).

Fetal liver erythroid precursors cells were cultured in StemPro-34 (ThermoFisher) with 1x nutrient supplement (ThermoFisher), 2 mM glutamax (ThermoFisher), 1% penicillin-streptomycin (Thermo-Fisher), 100 µM monothioglycerol (Sigma), 1 µM dexamethasone (Sigma), 0.5 U/ml of erythropoietin, and 1% conditioned medium from a kit ligand producing CHO cell line. Cells were cultured in a humidified incubator at 37°C (5% carbon dioxide).

### shRNA-mediated knockdown

The vectors expressing MiR-30 context *luciferase*, murine *Exosc8 (Rrp43)*, *Exosc9 (Rrp45)* shRNAs were described (*McIver et al., 2014*).

MiR-30 context sh*Exosc3* 1 sequence:
TGCTGTTGACAGTGAGCG**AACTGGCAGAGAGTTGACATAT**TAGTGAAGCCACAGATGTA**ATA TGTCAACTCTCTGCCAGTC**TGCCTACTGCCTCGGA.

MiR-30 context sh*Exosc3* 2 sequence:
TGCTGTTGACAGTGAGCG**CAAGACCATTCAGCAGACGTTA**TAGTGAAGCCACAGATGTA **TAACGTCTGCTGAATGGTCTTA**TGCCTACTGCCTCGGA.

Bold sequences denote sense and antisense sequences.

Wild type *Kit* expression vector was a kind gift from Ruben Kapur. G1E-ER-GATA-1 cells (5 x $10^5$), and primary murine fetal liver erythroid precursors (2 × $10^5$) were spinfected (*McIver et al., 2014*).

### Protein co-immunoprecipitation

48 hr post-infection 20 × $10^6$ G1E-ER-GATA1 cells were harvested and resuspended in 500 µl NP40 lysis buffer (50 mM Tris pH 8 (Sigma), 150 mM NaCl (Sigma), 5 mM DTT (Sigma), 0.5% NP40 (Sigma) supplemented with protease inhibitors (0.2mM PMSF (Sigma), 20µg/ml leupeptin (Roche, Basel, Switzerland)), 50 µg/ml RNase A (Sigma) and 2 µl/ml DNase I (ThermoFisher). Cells were lysed on ice for 30min and insoluble cell debris were removed by centrifugation. After taking an input sample (25 µl) the lysate was pre-cleared with 20 µl protein-A Agarose (Sigma) and 5 µl rabbit pre-immune sera for 1 hr at 4°C. After pelleting the protein-A Agarose, the supernatant was incubated with 5 µg rabbit IgG or 5 µg rabbit anti-Exosc3 (Abcam, San Francisco CA, Ab156683) overnight at 4°C, before addition of a 15 µl protein-A Agarose pellet for a further 2 hr. The protein-A Agarose pellet was washed once with NP40 lysis buffer and three times with TEG buffer (10 mM Tris pH 7.6, 50 mM NaCl, 4 mM EDTA (Sigma), 5 mM DTT, 10% Glycerol (Sigma)). The protein-A Agarose was resuspended in 50 µl of 2x SDS lysis buffer (25 mM Tris pH 6.8, 6% SDS (Sigma), 4% β-mercaptoethanol (Sigma), 10% glycerol, 0.02% bromophenol blue (Sigma)), 25 µl of 2x SDS lysis buffer was added to the input sample, and incubated at 100°C for 5 min. Proteins were resolved on an 11% SDS-PAGE gel and Exosc2 (Abcam ab156698) was measured by semi-quantitative Western blotting with ECL Plus (ThermoFisher).

### Quantitative real-time RT-PCR

Total RNA was purified with Trizol (ThermoFisher). cDNA was prepared by annealing 1 µg or 0.2 µg (sorted samples) of RNA with 250 ng of a 1:5 mixture of random hexamer and oligo (dT) primers (Eurofins, Louisville, KY) heated at 68°C for 10 min. This was followed by incubation with Murine Moloney Leukemia Virus Reverse Transcriptase (ThermoFisher) with 10 mM DTT, RNasin (Promega, Madison, Wi), and 0.5 mM dNTPs at 42°C for 1 hr. The mixture was diluted to a final volume of 100 µl and heat inactivated at 95°C for 5 min. cDNA was analyzed in reactions (20 µl) containing 2 µl of cDNA, appropriate primers (Eurofins), and 10 µl of SYBR green master mix (ThermoFisher). Product accumulation was monitored by SYBR green fluorescence. A standard curve of serial dilutions of cDNA samples was used to determine relative expression. mRNA levels were normalized to 18S rRNA. Primer sequences are found in *Supplementary file 1*.

## Flow cytometry and cell sorting

For quantitation of cell surface markers, $1 \times 10^6$ cells were stained in 100 µl PBS/10% FBS (Gemini) with anti-mouse Ter119-APC (RRID:AB_469474) (1:100), CD71-PE (RRID:AB_465741) (1:100) and Kit-PEcy7 (RRID: AB_469644) (1:100) (eBioscience, San Diego, CA), at 4°C for 30 min in the dark. To quantitate apoptosis after CD71/Ter119 staining, cells were washed in Annexin V Buffer (10 mM HEPES (Sigma), 140 mM NaCl, 2.5 mM $CaCl_2$ (Sigma), pH 7.4) then stained with Annexin V-Pacific blue (ThermoFisher) (1:40) and DRAQ7 (Abcam) (1:100) for 20 min in the dark at room temperature. To detect intracellular phosphorylated Akt, erythroid precursor cells were expanded for 48 hr and sorted into Ter119$^+$ and Ter119$^-$ populations using magnetic beads (StemCell Technologies). Cells were serum-starved in 1% BSA (Sigma) in IMDM for 1 hr at 37°C before stimulation with either 10 ng/ml SCF (Merk Millipore, Billerica, MA) or 2 U/ml Epo (Amgen) for 10 min and fixed in 2% paraformaldehyde (Alfa Aesar, Ward Hill, MA) for 10 min at 37°C. After permeabilization overnight at -20°C in 95% methanol (ThermoFisher) cells were incubated for 1 hr in HBSS (ThermoFisher)/4% FBS at 4°C. Cells were stained with rabbit phospho-Akt or rabbit phospho-ERK (1:200) (Cell Signaling, Danvers, MA) for 30 min before incubation in goat anti rabbit-APC (1:200) (Jackson ImmunoResearch, West Grove, PA), Kit-PEcy7 (1:100) and CD71-PE (1:100) for 30 min at room temperature. To analyze cell cycle of erythroid precursor cells, fetal liver erythroid precursors were expanded for 24 or 72 hr, and Ter119$^-$ cells were isolated. Cells were fixed/permeabilized in 70% ethanol and stained with 5 µg/ml DAPI (Biolegend, San Diego, CA) overnight at -20°C. Cells were washed twice in PBS before analysis. Samples were analyzed using a BD LSR II (BD Biosciences, San Jose, CA) or sorted into distinct populations using a BD FACSAria II. DAPI or DRAQ7 (Abcam) were used for apoptotic analyses, and for fixed cells, Zombie UV (Biolegend) staining discriminated dead cells.

## Colony assay

R1 cells were FACS-sorted 24 hr post-*Exosc8*-knockdown, and 5000 cells were plated in duplicate in Methocult M3434 (StemCell Technologies) according to the manufacturer's instructions. CFU-E and BFU-E colonies were quantitated after culturing for 2 and 8 days, respectively, at 37°C with 5% $CO_2$

## Total protein analysis

24 hr post-*Exosc8* knockdown, Ter119$^+$ cells were depleted from the primary erythroid precursor samples. Equal numbers of cells were boiled for 10 min in SDS lysis buffer. Proteins were resolved by SDS-PAGE and incubated with rabbit anti-Kit (Cell Signaling, D13A2 RRID:AB_1147633), rabbit anti-GATA-2 or rat anti-GATA-1 (Santa Cruz, Dallas TX, sc-265 RRID:AB_627663).

## Quantitative chromatin immunoprecipitation (ChIP) assay

Primary fetal liver erythroid precursor cells ($2 \times 10^6$) were crosslinked with 1% formaldehyde (Sigma) for 10 min. Lysates were immunoprecipitated with antibodies against phospho-Ser5 Pol II (Covance, Princeton, NJ, H14 MMS-134R RRID:AB_10063994) or GATA-1 (*Grass et al., 2006*) using rabbit pre-immune serum or control IgM as control. For Exsoc9 ChIP, cells ($8 \times 10^6$) were crosslinked, and lysates were immunoprecipitated with anti-Exosc9 antibody (Novus Biologicals, Littleton, CO, NBP1-71702 RRID:AB_11026964) using control IgG as control. DNA was quantitated by real-time PCR with SYBR green fluorescence. Primers sequences used to assess protein occupancy are indicated in *Supplementary file 2*.

## Statistics

A Students T-test was used to compare experimental and control samples. When comparing multiple groups, ANOVA was conducted to identify any significant variance between samples, followed by a Tukey-Kramer test to identify statistical relationships between each pair of samples within the experiment. All analysis was conducted using JMP software (SAS Institute Inc. Cary, NC). Asterisks indicate significance relative to control, *p<0.05, **p<0.01 and ***p<0.001.

## Acknowledgements

We thank R Kapur for providing the wild type *Kit* expression vector and MJ Weiss for providing retroviral vectors and the cell culture protocol. This work was supported by grants HL116365 and

DK50107 to EHB and Cancer Center Support Grant P30 CA014520. SCM was supported by an American Heart Association postdoctoral fellowship. The authors have no conflict of interest to disclose.

## Additional information

### Funding

| Funder | Grant reference number | Author |
|---|---|---|
| American Heart Association | Postdoctoral fellowship | Skye C McIver |
| National Institutes of Health | DK50107 | Emery H Bresnick |
| National Institutes of Health | HL116365 | Emery H Bresnick |
| University of Wisconsin Carbone Cancer Center | Cancer Center Support Grant (P30 CA014520) | Emery H Bresnick |

The funders had no role in study design, data collection and interpretation, or the decision to submit the work for publication.

### Author contributions

SCM, Conception and design, Acquisition of data, Analysis and interpretation of data, Drafting or revising the article; KRK, Acquisition of data, Analysis and interpretation of data, Drafting or revising the article; ED, PL, DY, Acquisition of data, Analysis and interpretation of data; Y-AK, Conception and design, Drafting or revising the article; EHB, Conception and design, Analysis and interpretation of data, Drafting or revising the article

### Author ORCIDs

Emery H Bresnick, http://orcid.org/0000-0002-1151-5654

### Ethics

This study was performed in strict accordance with the recommendations in the Guide for the Care and Use of Laboratory Animals of the National Institutes of Health. All of the animals were handled according to approved institutional animal care and use committee (IACUC) protocols (#M02230) of the University of Wisconsin-Madison.

## Additional files

### Supplementary files

• Supplementary file 1. Primers used for analysis of mRNA expression by qRT-PCR.

• Supplementary file 2. Primers used for analysis of protein occupancy by qChIP.

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
