## [Decision Letter]

Thank you for submitting your article "Orchestrating Developmental Signaling to Balance Self-Renewal and Differentiation" for consideration by *eLife*. Your article has been reviewed by three peer reviewers, one of whom, Amy Wagers, is a member of our Board of Reviewing Editors, and the evaluation has been overseen by Sean Morrison as the Senior Editor.

The reviewers have discussed the reviews with one another and the Reviewing Editor has drafted this decision to help you prepare a revised submission.

Summary:

The manuscript by Bresnick and colleagues investigates the role of exosome complex proteins in erythroid differentiation. Exosome complexes are highly conserved RNA processing/degrading machines that are present in almost all eukaryotic cells. Here, the authors explore the role of individual exosome subunits in complex stability and report that disruption of EXOSC8 or 9 perturbs exosome complex assembly, prevents normal signaling by the c-kit receptor and accelerates Epo responsiveness, leading to enhanced erythroid maturation. They further dissect likely mechanisms by which exosome complex disruption alters erythropoiesis, and implicate transcriptional control of c-kit as a key regulatory mechanism. Overall, the study is well-designed and executed and the data are presented clearly and provide compelling support for the authors' conclusions. Although investigation of the in vivo relevance of exosome complex activity to the generation of erythroid cells is notably lacking, in light of the fact that exosome complex functions in stem/progenitor cells have been relatively unexplored to date, the in vitro results reported here provide an appropriate model system to begin exploring this biology and yield new insights into potential regulatory mechanisms that dictate cell fate decisions during differentiation of mammalian cells. However, a major weakness of the paper is its lack of mechanistic insight into EXOSC8/9 mediated control of Kit expression. Additional experiments are needed to clarify the mechanism by which loss of Exosc8/9 leads to a loss of Kit expression beyond what the authors have previously published (McIver et al. (2014) Blood 124:2285) and to warrant publication in *eLife*.

Essential revisions:

1) Currently, the link between exosome complex integrity and c-kit expression is unclear (Figure 7). The authors should perform further studies to provide at least some insight into how exosome disassembly leads to altered expression of c-kit. There are several possibilities as the authors pointed out in the discussion, but none of them were investigated. In particular:

1A) The authors propose that Exosc8/9 inhibits maturation of erythroid cells through maintaining the transcription of Kit. They demonstrate this by showing that Exosc8 knockdown caused reduced phospho-Ser5 Pol II occupancy within the coding region and 3' UTR of the Kit gene. Using phospho Pol II binding does not demonstrate a direct mechanism of Exosc8 action since the loss of Kit transcription could be regulated by other indirect mechanisms. Furthermore, in the authors' previous paper they demonstrated that loss of Exosc8 resulted in increased phospho-Ser5 Pol II at the promoters of *Hbb-b1, Alas2*, and *Slc4a1* which resulted in their increased mRNA levels. How does Exosc8 in certain contexts act as an activator of transcription while in other contexts it acts as an inhibitor? The authors should test the legitimacy of direct mechanism of action for Exosc8 on these genes. Does Exosc8 bind to the Kit gene, whether through the promoter or coding sequence? If it does bind, is there a particular sequence that it binds?

1B) The authors propose an attractive hypothesis that exosome complex mediated regulation of RNA maintains progenitor cells in an undifferentiated state allowing for expansion of the progenitor population prior to differentiation, but their data are not sufficient to support this idea. Is Kit the only target? W mutants do not exhibit precocious differentiation. Epo signaling as measured by Akt-P and Erk-P is unaffected, so one question is whether the exosome alters the expression of a factor necessary for Epo dependent differentiation? Does the transcriptome data from the previous paper point to any other genes involved in this process? Work from the Socolovsky lab showed that cell cycle changes are key aspects of the switch from proliferation to erythroid differentiation, and the authors' own previous work showed increases in cell cycle regulators when exosome complex components were knocked down. Is this loss of Kit expression tied to cell cycle changes?

2) The Kit overexpression rescue experiment is hard to interpret because expression of Kit itself seems to block erythroid maturation. Kit expression seems to supershift the cells to an R2 fate and inhibit the R3 phenotype in both control and Exosc8 knockdown cells making it hard to discern whether Kit is truly rescuing the Exosc8 knockdown phenotype or merely shifting everything to an R2 fate regardless of the Exosc8 status. The authors should quantify the overexpression of Kit by flow cytometry or Western and ensure that Kit expression in Exosc8 knockdown cells is only expressed to endogenous control levels and not vastly overexpressed to cause everything to be supershifted to an R2 fate. Figure 6 shows that Kit is overexpressed by 5-7.5 fold its normal levels. Thus, the rescue experiment in Exosc8 knockdown cells needs to be done with levels of Kit that are at endogenous control levels.

3) One of the questions in the exosome field in regards to regulation of development/differentiation is whether the exosome acts as subcomplexes or whether the entire exosome is necessary for function. As the authors cited, work in *Arabidopsis* and *Drosophila* has demonstrated that each exosome subunit has seemingly distinct functions. The authors attempt to address this by demonstrating that without Exosc8 or Exosc9, the stability of the complex is compromised due to a loss of interaction between Exosc3 and Exosc2. However, this is not a robust measure of complex integrity. The authors should knockdown Exosc8 or Exosc9 and then probe for Exosc3 or Exosc2 using a native gel to determine the size of the complex before and after subunit knockdown. Furthermore it is important to determine whether Exosc3 and Exosc2 are necessary for erythroid cell self-renewal. The authors can do this by knocking down Exosc3 and Exosc2 and evaluating the effects on erythroid maturation, similar to the Exosc8 or Exosc9 knockdowns.

4) The authors consistently refer to the role of the exosome complex in self-renewal, which is a tricky term. Self-renewal is best understood in the context of stem cells, where it refers to the maintenance of stem cells. For example during an asymmetric division one progeny is another stem cell maintaining the population (self-renewal), while the other progeny is not a stem cell (differentiation). The Lin- population of fetal liver cells is not a pure population of cells so it is difficult to conclude that Kit signaling maintains self-renewal of this population. A better term would be proliferation of a transient amplifying population.

5) The authors should explicitly discuss the need for in vivo verification of the model they propose as a future direction of their research.

[Editors' note: further revisions were requested prior to acceptance, as described below.]

Thank you for resubmitting your work entitled "Orchestrating Developmental Signaling to Balance Proliferation and Differentiation" for further consideration at *eLife*. Your revised article has been favorably evaluated by Sean Morrison as the Senior editor and Amy Wagers as the Reviewing editor.

We appreciate your responsiveness to the prior review comments and while the manuscript has been improved, there are some remaining issues that need to be addressed before acceptance, as outlined below:

1) The existing title is too vague. It is important to include reference to the system and mechanism being studied. Perhaps something like: Role of the exosome complex in orchestrating developmental signaling to balance proliferation and differentiation during erythropoiesis.

2) The insertion point for the description of new data on Exosc3 inhibition causes the text to be a bit disjointed. I suggest moving the sentences to immediately following the first sentence. The paragraph would then read more smoothly. See suggested edits below:

"Previously, we demonstrated that downregulating exosome complex subunits (Exosc8, Exosc9 or Dis3) in murine fetal liver erythroid precursor cells induced erythroid maturation (McIver et al., 2014). Analogous to Exosc8 and Exosc9 (McIver et al., 2014), experiments in which Exosc3 expression is impaired suggest that this protein also suppresses maturation of primary murine fetal liver lineage-negative hematopoietic precursor cells. In particular, Exosc3 downregulation using two distinct shRNAs increased the R4 (late basophilic/orthochromatic erythroblasts) cell population 9 fold (p = 0.006 and p = 0.01 for the two shRNAs, respectively) (Figure 1—figure supplement 1). However, it remains unclear whether the single subunit perturbations impact exosome complex integrity. To address this, we developed a co-immunoprecipitation assay to test whether individual components mediate complex integrity (Figure 1). Using the X-ray crystal structure of the exosome complex as a guide (Liu et al., 2006), the strategy involved testing whether downregulating endogenous Exosc8 or Exosc9 alter interactions between endogenous Exosc2 and Exosc3, subunits that do not interact directly in the complex (Figure 1). As Exosc2 and Exosc3 are only expected to co-immunoprecipitate when residing in the complex or a sub-complex, the extent of co-immunoprecipitation constitutes a metric of complex integrity."

3) In the Discussion, the authors have used the terms "self-renewal/proliferation" and "stem/progenitor" several times. While this term is certainly used by many others in the literature, it is rather ambiguous, and could be confusing to those who are new to the field. Since *eLife* aims to reach a very broad audience, if it is possible to be more precise in any of these uses, the authors should do so, as it will improve the clarity and accessibility of their discussion. If it is not possible to be more precise, e.g., because the data available do not allow for unambiguous discrimination of the process or cell type involved, then it is reasonable to retain the hybrid term.

4) While the authors' additional experiments provide stronger support for the role of kit downregulation in the accelerated maturation phenotype they observe with Exocs8 manipulation, it would be helpful to include a few sentences in the Discussion related to the relationship of the current results to phenotypes in Kit-mutant mice (similar to the response letter), in order to proactively address this consideration for other readers for whom the data may seem somewhat confusing.

---

## [Author Response]

*Essential revisions:*

*1) Currently, the link between exosome complex integrity and c-kit expression is unclear (Figure 7). The authors should perform further studies to provide at least some insight into how exosome disassembly leads to altered expression of c-kit. There are several possibilities as the authors pointed out in the discussion, but none of them were investigated. In particular:*

*1A) The authors propose that Exosc8/9 inhibits maturation of erythroid cells through maintaining the transcription of Kit. They demonstrate this by showing that Exosc8 knockdown caused reduced phospho-Ser5 Pol II occupancy within the coding region and 3' UTR of the Kit gene. Using phospho Pol II binding does not demonstrate a direct mechanism of Exosc8 action since the loss of Kit transcription could be regulated by other indirect mechanisms. Furthermore, in the authors' previous paper they demonstrated that loss of Exosc8 resulted in increased phospho-Ser5 Pol II at the promoters of Hbb-B1, Alas2, and Slc4a1 which resulted in their increased mRNA levels. How does Exosc8 in certain contexts act as an activator of transcription while in other contexts it acts as an inhibitor? The authors should test the legitimacy of direct mechanism of action for Exosc8 on these genes. Does Exosc8 bind to the Kit gene, whether through the promoter or coding sequence? If it does bind, is there a particular sequence that it binds?*

We appreciate the reviewers’ instructive comments. In new experimentation, we used qChIP to test whether endogenous Exosc9 occupies various regions of *Kit* (Figure 6, Results section, subsection “Mechanism governing exosome complex-regulated developmental singaling: transcription induction of an essential receptor tyrosine kinase”) and also promoters of genes previously identified to be upregulated by exosome complex disruption. Exosc9 occupied the *Kit* promoter and open reading frame, as well as *Alas2, Slc4a1* and *Hbb-b1* promoters. As discussed in the text, the exosome complex can associate with elongating RNA Polymerase II and directly suppresses synthesis of PROMPTs at promoters. Thus, our results are consistent with expectations that the complex would have a reasonably broad distribution at endogenous loci, rather than residing at a specific site, like a transcription factor with intrinsic sequence-specific DNA binding activity. Regarding why the exosome complex might function as a transcriptional activator and repressor, many transcriptional factors and coregulators mediate activation or repression in a context-dependent manner. It will be important to dissect how the exosome complex might toggle between activation and repression states and/or how the chromosomal locus dictates exosome complex functionality. Addressing this issue will require a major new project beyond the scope of our current discovery of an exosome complex-Kit regulatory mechanism that orchestrates a developmental signaling switch that determines erythroid precursor proliferation versus differentiation.

*1B) The authors propose an attractive hypothesis that exosome complex mediated regulation of RNA maintains progenitor cells in an undifferentiated state allowing for expansion of the progenitor population prior to differentiation, but their data are not sufficient to support this idea. Is Kit the only target? W mutants do not exhibit precocious differentiation. Epo signaling as measured by Akt-P and Erk-P is unaffected, so one question is whether the exosome alters the expression of a factor necessary for Epo dependent differentiation? Does the transcriptome data from the previous paper point to any other genes involved in this process? Work from the Socolovsky lab showed that cell cycle changes are key aspects of the switch from proliferation to erythroid differentiation, and the authors' own previous work showed increases in cell cycle regulators when exosome complex components were knocked down. Is this loss of Kit expression tied to cell cycle changes?*

Mice homozygous for the W mutation (white spotting) are sterile and die perinatally form severe macrocytic anemia, which can be rescued by ectopic expression of Epo (Waskow, et al. 2004). Macrocytic anemia of W mice is apparent by E12, persists after birth and is associated with much lower red cell numbers, presumably caused by inefficient proliferation of hematopoietic precursor cells (Russel et al. 1968). Anemia in W mutants can be cured by transplant of hematopoietic tissue from wild type mice. Hematopoietic cells from W/W bone marrow exhibit a 200 fold decreased ability to form splenic colonies after transplant into irradiated recipients, indicating a deficiency in colony forming cells (McCulloch et al. 1964). In vitro colony assays demonstrated that BFU-E and CFU-E activity is reduced in bone marrow and spleen cells from W/W mice in comparison with wild type controls (Gregory and Eaves, 1978). Cells isolated from W/W fetal livers do not form splenic colonies when introduced into irradiated host animals. In in vitro colony assays, W/W fetal livers exhibit normal numbers of CFU-GM and BFU-E colonies, but generate much fewer CFU-Es (Nocka et al. 1989). These phenotypes reflect, at least in part, a defect in the formation and or function of both hematopoietic stem and progenitor cells. Importantly, our analyses involve isolation of wild type fetal liver hematopoietic precursors, which are cultured under conditions that support erythroid precursors. Our loss-of-function analysis demonstrated that Exosc8 downregulation reduces *Kit* expression in erythroid precursors. This is a very different scenario than analyzing Kit-defective HSCs and/or other multi-potent precursors. Thus, we do not believe our results are predictable or inconsistent with descriptions of W mutant phenotypes.

We conducted three lines of new experimentation to further address whether loss of Kit expression is a primary consequence of exosome complex disruption or an indirect consequence of cell cycle changes and/or erythroid maturation. We analyzed expression of *Exosc8, Kit* and GATA-1/Exosc8 regulated genes implicated in cell cycle arrest 24 h after infection with shRNA-expressing retroviruses. Whereas expression of both *Kit* and *Exosc8* were significantly reduced, the cell cycle-regulatory genes were unaffected at this time; previously, we analyzed expression of these genes 72 h post-knockdown (Figure 5, Results section, subsection “Mechanism governing exosome complex-regulated developmental singaling: transcription induction of an essential receptor tyrosine kinase”). Thus, reduced *Kit* expression occurs before alterations in expression of cell cycle regulatory genes.

We analyzed the percentage of Kit-expressing Ter119-negative cells in G1, S and G2/M phases 24 h and 72 h post-infection. At 24 h post-*Exosc8* knockdown, there was only a slight increase (3.5%) in G1 cells. However, at 72 h post-*Exosc8* knockdown, G1 cells increased by 17% (Figure 5). Thus, *Kit* repression occurs prior to the prominent cell cycle arrest observed after *Exosc8*knockdown.

To eliminate the possibility that reduced *Kit* expression resulting from Exosc8 downregulation is a consequence of erythroid maturation, we analyzed *Kit* expression in *Exosc8*-knockdown G1E cells, a GATA-1 null proerythroblast cell line that is not competent to differentiate. At 48 h post-infection with retrovirus expressing *Exosc8* shRNA, *Kit* mRNA was reduced by 37% (Figure 5, page 13, lines 3-4). Kit expression on the cell surface was reduced by 44% (Figure 5). Thus, *Kit* repression resulting from *Exosc8* knockdown is not a consequence of, maturation and occurs prior to maturation resulting from exosome complex disruption.

*2) The Kit overexpression rescue experiment is hard to interpret because expression of Kit itself seems to block erythroid maturation. Kit expression seems to supershift the cells to an R2 fate and inhibit the R3 phenotype in both control and Exosc8 knockdown cells making it hard to discern whether Kit is truly rescuing the Exosc8 knockdown phenotype or merely shifting everything to an R2 fate regardless of the Exosc8 status. The authors should quantify the overexpression of Kit by flow cytometry or Western and ensure that Kit expression in Exosc8 knockdown cells is only expressed to endogenous control levels and not vastly overexpressed to cause everything to be supershifted to an R2 fate. Figure 6 shows that Kit is overexpressed by 5-7.5 fold its normal levels. Thus, the rescue experiment in Exosc8 knockdown cells needs to be done with levels of Kit that are at endogenous control levels.*

We appreciate the reviewers’ instructive comments. We conducted new experimentation to address this important issue. We repeated the Kit rescue experiment using a titration of Kit-expressing retrovirus. We established conditions in which enforced expression of Kit restores Kit cell surface expression in the *Exosc8* knockdown cells to the endogenous level. Importantly, this level of *Kit* expression negated the increase in the R3 population cells caused by *Exosc8* knockdown (Figure 7).

*3) One of the questions in the exosome field in regards to regulation of development/differentiation is whether the exosome acts as subcomplexes or whether the entire exosome is necessary for function. As the authors cited, work in Arabidopsis and Drosophila has demonstrated that each exosome subunit has seemingly distinct functions. The authors attempt to address this by demonstrating that without Exosc8 or Exosc9, the stability of the complex is compromised due to a loss of interaction between Exosc3 and Exosc2. However, this is not a robust measure of complex integrity. The authors should knockdown Exosc8 or Exosc9 and then probe for Exosc3 or Exosc2 using a native gel to determine the size of the complex before and after subunit knockdown. Furthermore it is important to determine whether Exosc3 and Exosc2 are necessary for erythroid cell self-renewal. The authors can do this by knocking down Exosc3 and Exosc2 and evaluating the effects on erythroid maturation, similar to the Exosc8 or Exosc9 knockdowns.*

We appreciate the reviewers’ instructive comments. We believe that our test of whether downregulating a single exosome complex subunit impacts a protein-protein interaction between two other endogenous subunits, as guided by the X-ray structure, is an effective (and unique) means of measuring complex integrity. There are no examples in which the exosome complex has been resolved on a nature protein gel. We have considerable biochemical expertise and predict that several months of developmental work would be required to assess feasibility of establishing this type of assay. Furthermore, mobility of a multimeric complex in a native gel will be influenced by *complex size and shape,* and therefore the consequences of disrupting interactions within the complex may not be predictable, based on simple assumptions. Based on this logic, we have not pursued this approach.

We conducted new experimentation to assess whether Exosc3 also suppresses erythroid maturation. We downregulated Exosc3 with two independent shRNAs and quantitated erythroid maturation. Exosc3 downregulation significantly increased mature erythroblasts (R4 population) (Figure 1—figure supplement 1 and first paragraph of Results section). Exosc3 resembles Exosc8 and Exosc9 in suppressing erythroid maturation, strongly supporting our model that the exosome complex establishes an erythroid maturation barricade.

*4) The authors consistently refer to the role of the exosome complex in self-renewal, which is a tricky term. Self-renewal is best understood in the context of stem cells, where it refers to the maintenance of stem cells. For example during an asymmetric division one progeny is another stem cell maintaining the population (self-renewal), while the other progeny is not a stem cell (differentiation). The Lin- population of fetal liver cells is not a pure population of cells so it is difficult to conclude that Kit signaling maintains self-renewal of this population. A better term would be proliferation of a transient amplifying population.*

We changed “self-renewal” to proliferation and/or proliferation/amplification when we refer to erythroid precursors.

*5) The authors should explicitly discuss the need for in vivo verification of the model they propose as a future direction of their research.*

We have added a statement in the Discussion.

[Editors' note: further revisions were requested prior to acceptance, as described below.]

*1) The existing title is too vague. It is important to include reference to the system and mechanism being studied. Perhaps something like: Role of the exosome complex in orchestrating developmental signaling to balance proliferation and differentiation during erythropoiesis.*

We revised the title as recommended.

*2) The insertion point for the description of new data on Exosc3 inhibition causes the text to be a bit disjointed. I suggest moving the sentences to immediately following the first sentence. The paragraph would then read more smoothly. See suggested edits below:*

*"Previously, we demonstrated that downregulating exosome complex subunits (Exosc8, Exosc9 or Dis3) in murine fetal liver erythroid precursor cells induced erythroid maturation (McIver et al., 2014). […] As Exosc2 and Exosc3 are only expected to co-immunoprecipitate when residing in the complex or a sub-complex, the extent of co-immunoprecipitation constitutes a metric of complex integrity."*

We revised the text in first paragraph of Reuslts section, exactly as recommended.

*3) In the Discussion, the authors have used the terms "self-renewal/proliferation" and "stem/progenitor" several times. While this term is certainly used by many others in the literature, it is rather ambiguous, and could be confusing to those who are new to the field. Since eLife aims to reach a very broad audience, if it is possible to be more precise in any of these uses, the authors should do so, as it will improve the clarity and accessibility of their discussion. If it is not possible to be more precise, e.g., because the data available do not allow for unambiguous discrimination of the process or cell type involved, then it is reasonable to retain the hybrid term.*

In the Discussion, we eliminated the hybrid terms self-renewal/proliferation and stem/progenitor.

*4) While the authors' additional experiments provide stronger support for the role of kit downregulation in the accelerated maturation phenotype they observe with Exocs8 manipulation, it would be helpful to include a few sentences in the Discussion related to the relationship of the current results to phenotypes in Kit-mutant mice (similar to the response letter), in order to proactively address this consideration for other readers for whom the data may seem somewhat confusing.*

We added a short paragraph in fourth paragraph of Discussion to describe the prior work on W mutant mice, as we had articulated in the prior letter addressing reviewers’ comments.